# CERTIFIED TRAINING WITH BRANCH-AND-BOUND: A CASE STUDY ON LYAPUNOV-STABLE NEURAL CONTROL

## ABSTRACT

We study the problem of learning Lyapunov-stable neural controllers which provably satisfy the Lyapunov asymptotic stability condition within a region-of-attraction. Compared to previous works which commonly used counterexample guided training on this task, we develop a new and generally formulated certified training framework named **CT-BaB**, and we optimize for differentiable verified bounds, to produce verification-friendly models. In order to handle the relatively large region-of-interest, we propose a novel framework of training-time branch-and-bound to dynamically maintain a training dataset of subregions throughout training, such that the hardest subregions are iteratively split into smaller ones whose verified bounds can be computed more tightly to ease the training. We demonstrate that our new training framework can produce models which can be more efficiently verified at test time. On the largest 2D quadrotor dynamical system, verification for our model is more than 5X faster compared to the baseline, while our size of region-of-attraction is 16X larger than the baseline.

## 1 INTRODUCTION

Deep learning techniques with neural networks (NNs) have greatly advanced abundant domains in recent years. Despite the impressive capability of NNs, it remains challenging to obtain provable guarantees on the behaviors of NNs, which is critical for the trustworthy deployment of NNs especially in safety-critical domains. One area of particular concern is safe control for robotic systems with NN-based controllers (Chang et al., 2019; Dai et al., 2021; Wu et al., 2023; Yang et al., 2024). There are many desirable properties in safe control, such as reachability w.r.t. target and avoid sets (Althoff & Kochdumper, 2016; Bansal et al., 2017; Dutta et al., 2019; Everett et al., 2021; Wang et al., 2023b), forward invariance (Ames et al., 2016; Taylor et al., 2020; Zhao et al., 2021; Wang et al., 2023a; Huang et al., 2023), stability (Lyapunov, 1992; Chang et al., 2019; Dai et al., 2021; Wu et al., 2023; Yang et al., 2024), etc.

In particular, we focus on the Lyapunov (Lyapunov, 1992) asymptotic stability of NN-based controllers in discrete-time nonlinear dynamical systems (Wu et al., 2023; Yang et al., 2024), where we aim to train and verify asymptotically Lyapunov-stable NN-based controllers. It involves training a controller while also finding a Lyapunov function which intuitively characterizes the energy of input states in the dynamical system, where the global minima of the Lyapunov function is at an equilibrium state. If it can be guaranteed that for any state within a region-of-attraction (ROA), the controller always makes the system evolve towards states with lower Lyapunov function values, then it implies that starting from any state within the ROA, the controller can always make the system converge towards the equilibrium state and thus the stability can be guaranteed. Such stability requirements have been formulated as the Lyapunov condition in the literature. This guarantee is for an infinite time horizon and implies a convergence towards the equilibrium, and thus it is relatively stronger than reachability or forward invariance guarantees.

Previous works (Wu et al., 2023; Yang et al., 2024) typically used a counterexample-guided procedure that basically tries to find concrete inputs which violate the Lyapunov condition and then train models on counterexamples. After the training, the Lyapunov condition is verified by a formal verifier for NNs (Zhang et al., 2018; Xu et al., 2020; 2021; Wang et al., 2021; Zhang et al., 2022;

Shi et al., 2024). However, the training process has very limited consideration on the computation of verification which is typically achieved by computing verified output bounds given an input region. Thereby, their models are often not sufficiently "verification-friendly", and the verification can be challenging and take a long time after training (Yang et al., 2024).

In this paper, we propose to consider the computation of verification during the training, for the first time on the problem of learning Lyapunov-stable neural controllers. To do this, we optimize for *verified bounds on subregions of inputs* instead of only violations on concrete counterexample data points, and thus our approach differs significantly compared to Wu et al. (2023); Yang et al. (2024). Optimizing for verified bounds during training is also known as "certified training" which was originally proposed for training provably robust NNs (Wong & Kolter, 2018; Mirman et al., 2018; Gowal et al., 2018; Müller et al., 2022; Shi et al., 2021; De Palma et al., 2022; Mao et al., 2024) under adversarial robustness settings (Szegedy et al., 2014; Goodfellow et al., 2015). However, our certified training here is significantly different, as we require that the model should globally satisfy desired properties on an entire large region-of-interest over the input space, rather than only local robustness guarantees around a finite number of data points. Additionally, the model in this problem contains not only an NN as the controller, but also a Lyapunov function and nonlinear operators from the system dynamics, introducing additional difficulty to the training and verification.

We propose a new **C**ertified **T**raining framework enhanced with training-time **B**ranch-**a**nd-**B**ound, namely **CT-BaB**. We jointly train a NN controller and a Lyapunov function by computing and optimizing for the verified bounds on the violation of the Lyapunov condition. To achieve certified guarantees on the entire region-of-interest, we dynamically maintain a training dataset which consists of subregions in the region-of-interest. We split hard examples of subregions in the dataset into smaller ones during the training, along the input dimension where a split can yield the best improvement on the training objective, so that the training can be eased with tighter verified bounds for the smaller new subregions. Our new certified training framework is generally formulated for problems requiring guarantees on an entire input region-of-interest, but we focus on the particular problem of learning Lyapunov-stable controllers in this paper as a case study.

Our work makes the following contributions:

- We propose a new certified training framework for producing NNs with relatively global guarantees which provably hold on the entire input region-of-interest instead of only small local regions around a finite number of data points. We resolve challenges in certified training for the relatively large input region-of-interest by proposing a training-time branch-and-bound method with a dynamically maintained training dataset.

- We demonstrate the new certified training framework on the problem of learning (asymptotically) Lyapunov-stable neural controller. To the best of our knowledge, this is also the first certified training work for the task. Our new approach greatly reduced the training challenges observed in previous work. For example, unlike previous works (Chang et al., 2019; Wu et al., 2023; Yang et al., 2024) which required a special initialization from a linear quadratic regulator (LQR) during counterexample-guided training, our certified training approach works well by training from scratch with random initialization.

- We empirically show that our training framework produces neural controllers which verifiably satisfy the Lyapunov condition, with a larger region-of-attraction (ROA), and the Lyapunov condition can be much more efficiently verified at test time. On the largest 2D quadrotor dynamical system, we reduce the verification time from 1.1 hours (Yang et al., 2024) to 11.5 minutes, while our ROA size is 16X larger.

## 2 RELATED WORK

**Learning Lyapunov-stable neural controllers.** On the problem of learning (asymptotically) Lyapunov-stable neural controllers, compared to methods using linear quadratic regulator (LQR) or sum-of-squares (SOS) (Parrilo, 2000; Tedrake et al., 2010; Majumdar et al., 2013; Yang et al., 2023; Dai & Permenter, 2023) to synthesize linear or polynomial controllers with Lyapunov stability guarantees (Lyapunov, 1992), NN-based controllers have recently shown great potential in scaling to more complicated systems with larger region-of-attraction. Some works used sampled data points to synthesize empirically stable neural controllers (Jin et al., 2020; Sun & Wu, 2021; Dawson et al.,

2022; Liu et al., 2023) but they did not provide formal guarantees. Among them, although Jin et al. (2020) theoretically considered verification, they assumed an existence of some Lipschitz constant which was not actually computed, and they only evaluated a finite number of data points without a formal verification.

To learn neural controllers with formal guarantees, many previous works used a Counter Example Guided Inductive Synthesis (CEGIS) framework by iteratively searching for counterexamples which violate the Lyapunov condition and then optimizing their models using the counterexamples, where counterexamples are generated by Satisfiable Modulo Theories (SMT) solvers (Gao et al., 2013; De Moura & Bjørner, 2008; Chang et al., 2019; Abate et al., 2020), Mixed Integer Programming solvers (Dai et al., 2021; Chen et al., 2021; Wu et al., 2023), or projected gradient descent (PGD) (Madry et al., 2018; Wu et al., 2023; Yang et al., 2024). Among these works, Wu et al. (2023) has also leveraged a formal verifier (Xu et al., 2020) only to guarantee that the Lyapunov function is positive definite (which can also be achieved by construction as done in Yang et al. (2024)) but not other more challenging parts of the Lyapunov condition; Yang et al. (2024) used $\alpha,\beta$-CROWN (Zhang et al., 2018; Xu et al., 2020; 2021; Wang et al., 2021; Zhang et al., 2022; Shi et al., 2024) to verify trained models without using verified bounds for training. In contrast to those previous works, we propose to conduct certified training by optimizing for differentiable verified bounds at training time, where the verified bounds are computed for input subregions rather than violations on individual counterexample points, to produce more verification-friendly models.

**Verification for neural controllers on other safety properties.** Apart from Lyapunov asymptotic stability, there are many previous works on verifying other safety properties of neural controllers. Many works studied the reachability of neural controllers to verify the reachable sets of neural controllers and avoid reaching unsafe states (Althoff & Kochdumper, 2016; Dutta et al., 2019; Tran et al., 2020; Hu et al., 2020; Everett et al., 2021; Ivanov et al., 2021; Huang et al., 2022; Wang et al., 2023b; Schilling et al., 2022; Kochdumper et al., 2023; Jafarpour et al., 2023; 2024; Teuber et al., 2024). Additionally, many other works studied the forward invariance and barrier functions of neural controllers (Zhao et al., 2021; Wang et al., 2023a; Huang et al., 2023; Harapanahalli & Coogan, 2024; Hu et al., 2024; Wang et al., 2024). In contrast to the safety properties studied in those works, the Lyapunov asymptotic stability we study is a stronger guarantee which implies a convergence towards an equilibrium point, which is not guaranteed by reachability or forward invariance alone.

**NN verification and certified training.** On the general problem of verifying NN-based models on various properties, many techniques and tools have been developed in recent years, such as $\alpha,\beta$-CROWN (Zhang et al., 2018; Xu et al., 2020; 2021; Wang et al., 2021; Zhang et al., 2022; Shi et al., 2024), nnenum (Bak, 2021), NNV (Tran et al., 2020; Lopez et al., 2023), MN-BaB (Ferrari et al., 2021), Marabou (Wu et al., 2024), NeuralSAT (Duong et al., 2024), VeriNet (Henriksen & Lomuscio, 2020), etc. One technique commonly used in the existing NN verifiers is linear relaxation-based bound propagation (Zhang et al., 2018; Wong & Kolter, 2018; Singh et al., 2019), which essentially relaxes nonlinear operators in the model by linear lower and upper bounds and then propagates linear bounds through the model to eventually produce a verified bound on the output of the model. Verified bounds computed in this way are differentiable and thus have also been leveraged in certified training (Zhang et al., 2020; Xu et al., 2020). Some other certified training works (Gowal et al., 2018; Mirman et al., 2018; Shi et al., 2021; Müller et al., 2022; De Palma et al., 2022) used even cheaper verified bounds computed by Interval Bound Propagation (IBP) which only propagates more simple interval bounds rather than linear bounds. However, existing certified training works commonly focused on adversarial robustness for individual data points with small local perturbations. In contrast, we consider a certified training beyond adversarial robustness, where we aim to achieve a relatively global guarantee which provably holds within the entire input region-of-interest rather than only around a proportion of individual examples.

Moreover, since verified bounds computed with linear relaxation can often be loose, many of the aforementioned verifiers for trained models also contain a branch-and-bound strategy (Bunel et al., 2020; Wang et al., 2021) to branch the original verification problem into subproblems with smaller input or intermediate bounds, so that the verifier can more tightly bound the output. In this work, we explore a novel use of the branch-and-bound concept in certified training, by dynamically expanding

a training dataset and gradually splitting hard examples into smaller ones during the training, to enable certified training which eventually works for the entire input region-of-interest.

## 3 METHODOLOGY

### 3.1 PROBLEM SETTINGS

**Certified training problem.**   Suppose the input region-of-interest of the problem is defined by $\mathcal{B} \subseteq \mathbb{R}^d$ for input dimension $d$, and in particular, we assume $\mathcal{B}$ is an axis-aligned bounding box $\mathcal{B} = \{\mathbf{x} \mid \underline{\mathbf{b}} \leq \mathbf{x} \leq \overline{\mathbf{b}}, \mathbf{x} \in \mathbb{R}^d\}$ with boundary defined by $\underline{\mathbf{b}}, \overline{\mathbf{b}} \in \mathbb{R}^d$ (we use "$\leq$" for vectors to denote that the "$\leq$" relation holds for all the dimensions in the vectors). We define a model (or a computational graph) $g_{\boldsymbol{\theta}} : \mathbb{R}^d \to \mathbb{R}$ parameterized by $\boldsymbol{\theta}$, where $g_{\boldsymbol{\theta}}$ generally consists of one or more NNs and also additional operators which define the properties we want to certify (such as the Lyapunov condition in this work). The goal of certified training is to optimize for parameters $\boldsymbol{\theta}$ such that the following can be provably verified (we may omit $\boldsymbol{\theta}$ in the remaining part of the paper):

$$\forall \mathbf{x} \in \mathcal{B}, \ g_{\boldsymbol{\theta}}(\mathbf{x}) \leq 0, \tag{1}$$

where any $g_{\boldsymbol{\theta}}(\mathbf{x}) > 0$ can be viewed as a violation. Unlike previous certified training works (Gowal et al., 2018; Mirman et al., 2018; Zhang et al., 2020; Müller et al., 2022) which only considered certified adversarial robustness guarantees on small local regions as $\{\mathbf{x} : \|\mathbf{x} - \mathbf{x}_0\| \leq \epsilon\}$ around a finite number of examples $\mathbf{x}_0 \in \mathcal{B}$ in the dataset, we require Eq. (1) to be fully certified for any $\mathbf{x} \in \mathcal{B}$.

Neural network verifiers typically verify Eq. (1) by computing a provable upper bound $\overline{g}$ such that $\overline{g} \geq g(\mathbf{x})$ ($\forall \mathbf{x} \in \mathcal{B}$) provably holds, and Eq. (1) is considered as verified if $\overline{g} \leq 0$. For models trained without certified training, the upper bound computed by verifiers is usually loose, or it requires a significant amount of time to further optimize the bounds or gradually tighten the bounds by branch-and-bound at test time. Certified training essentially optimizes for objectives which take the computation of verified bounds into consideration, so that Eq. (1) not only empirically holds for any worst-case data point $\mathbf{x}$ which can be empirically found to maximize $g(\mathbf{x})$, but also the model becomes more verification-friendly, i.e., verified bounds become tighter and thereby it is easier to verify $\overline{g} \leq 0$ with less branch-and-bound at test time.

**Specifications for Lyapunov-stable neural control.**   In this work, we particularly focus on the problem of learning a certifiably Lyapunov-stable neural state-feedback controller with continuous control actions in a nonlinear discrete-time dynamical system, with asymptotic stability guarantees. We adopt the formulation from Yang et al. (2024). Essentially, there is a nonlinear dynamical system

$$\mathbf{x}_{t+1} = f(\mathbf{x}_t, u_t(\mathbf{x}_t)), \tag{2}$$

which takes the state $\mathbf{x}_t \in \mathbb{R}^d$ at the current time step $t$ and a continuous control input $u_t(\mathbf{x}_t) \in \mathbb{R}^{n_u}$, and then the dynamical system determines the state at the next time step $t + 1$. The control input $u_t(\mathbf{x}_t)$ is generated by a controller which is a NN here. The state of the dynamical system is also the input of the certified training problem.

Lyapunov asymptotic stability can guarantee that if the system begins at any state $\mathbf{x} \in \mathcal{S}$ within a region-of-attraction (ROA) $\mathcal{S} \subseteq \mathcal{B}$, it will converge to a stable equilibrium state $\mathbf{x}^*$. Following previous works, we assume that the equilibrium state is known, which can be manually derived from the system dynamics for the systems we study. To certify the Lyapunov asymptotic stability, we need to learn a Lyapunov function $V(\mathbf{x}_t) : \mathbb{R}^d \to \mathbb{R}$, such that the Lyapunov condition provably holds for the dynamical system in Eq. (2):

$$\forall \mathbf{x}_t \neq \mathbf{x}^* \in \mathcal{S}, \ V(\mathbf{x}_t) > 0, \ V(\mathbf{x}_{t+1}) - V(\mathbf{x}_t) \leq -\kappa V(\mathbf{x}_t), \tag{3}$$

and $V(\mathbf{x}^*) = 0$, where $\kappa > 0$ is a constant which specifies the exponential stability convergence rate. This condition essentially guarantees that at each time step, the controller always make the system progress towards the next state with a lower Lyapunov function value, and thereby the system will ultimately reach $\mathbf{x}^*$ which has the lowest Lyapunov function value given $V(\mathbf{x}^*) = 0$. Following Yang et al. (2024), we guarantee $V(\mathbf{x}^*) = 0$ and $\forall \mathbf{x}_t \neq \mathbf{x} \in \mathbb{R}^d, V(\mathbf{x}_t) > 0$ by the construction of the Lyapunov function, as discussed in Section 3.4, and we specify the ROA using a sublevel set of $V$ as $\mathcal{S} := \{\mathbf{x} \in \mathcal{B} \mid V(\mathbf{x}) < \rho\}$ with sublevel set threshold $\rho$. Since ROA is now restricted to be a

subset of $\mathcal{B}$ and the verification will only focus on $\mathcal{B}$, we additionally need to ensure that the state at the next time step does not leave $\mathcal{B}$, i.e., $\mathbf{x}_{t+1} \in \mathcal{B}$.

Overall, we want to verify $g(\mathbf{x}_t) \leq 0$ for all $\mathbf{x}_t \in \mathcal{B}$, where $g(\mathbf{x}_t)$ is defined as:

$$g(\mathbf{x}_t) := \min \left\{ \rho - V(\mathbf{x}_t),\, \sigma(V(\mathbf{x}_{t+1}) - (1-\kappa)V(\mathbf{x}_t)) + \sum_{1 \leq i \leq d} \sigma([\mathbf{x}_{t+1}]_i - \overline{\mathbf{b}}_i) + \sigma(\underline{\mathbf{b}}_i - [\mathbf{x}_{t+1}]_i) \right\}, \tag{4}$$

where $\mathbf{x}_{t+1}$ is given by Eq. (2), and $\sigma(x) = \max\{x, 0\}$ is also known as ReLU. For the specification in Eq. (4), $\rho - V(\mathbf{x}_t)$ means that for a state which is provably out of the considered ROA as $V(\mathbf{x}_t) \geq \rho$, we do not have to verify Eq. (3) or $\mathbf{x}_{t+1} \in \mathcal{B}$, and it immediately satisfies $g(\mathbf{x}_t) \leq 0$; $\sigma(V(\mathbf{x}_{t+1}) - (1-\kappa)V(\mathbf{x}_t))$ is the violation on the $V(\mathbf{x}_{t+1}) - V(\mathbf{x}_t) \leq -\kappa V(\mathbf{x}_t)$ condition in Eq. (3); and the "$\sum_{1 \leq i \leq d}$" term in Eq. (4) denotes the violation on the $\mathbf{x}_{t+1} \in \mathcal{B}$ condition. Verifying Eq. (4) for all $\mathbf{x}_t \in \mathcal{B}$ guarantees the Lyapunov condition for any $\mathbf{x} \in \mathcal{S}$ in the ROA (Yang et al., 2024). In the training, we try to make $g(\mathbf{x}_t) \leq 0$ verifiable by optimizing the parameters in the neural controller $u_t$ and the Lyapunov function $V(\mathbf{x}_t)$.

## 3.2 TRAINING FRAMEWORK

As we are now considering a challenging setting, where we want to guarantee $g(\mathbf{x}) \leq 0$ on the entire input region-of-interest $\mathcal{B}$, directly computing a verified bound on the entire $\mathcal{B}$ can produce very loose bounds. Thus, we split $\mathcal{B}$ into smaller subregions, and we we maintain a dataset with $n$ examples $\mathbb{D} = \{(\underline{\mathbf{x}}^{(1)}, \overline{\mathbf{x}}^{(1)}), (\underline{\mathbf{x}}^{(2)}, \overline{\mathbf{x}}^{(2)}), \cdots, (\underline{\mathbf{x}}^{(n)}, \overline{\mathbf{x}}^{(n)})\}$, where each example $(\underline{\mathbf{x}}^{(k)}, \overline{\mathbf{x}}^{(k)})$ $(1 \leq k \leq n)$ is a subregion in $\mathcal{B}$, defined as a bounding box $\{\mathbf{x} : \mathbf{x} \in \mathbb{R}^d, \underline{\mathbf{x}}^{(k)} \leq \mathbf{x} \leq \overline{\mathbf{x}}^{(k)}\}$ with boundary $\underline{\mathbf{x}}^{(k)}$ and $\overline{\mathbf{x}}^{(k)}$, and all the examples in $\mathbb{D}$ cover $\mathcal{B}$ as $\bigcup_{(\underline{\mathbf{x}}, \overline{\mathbf{x}}) \in \mathbb{D}} (\underline{\mathbf{x}}, \overline{\mathbf{x}}) = \mathcal{B}$. We dynamically update and expand the dataset during the training by splitting hard examples into more examples with even smaller subregions, as we will introduce in Section 3.3.

During the training, for each training example $(\underline{\mathbf{x}}, \overline{\mathbf{x}})$, we compute a verified upper bound of $g(\mathbf{x})$ for all $\mathbf{x}$ $(\underline{\mathbf{x}} \leq \mathbf{x} \leq \overline{\mathbf{x}})$ within the subregion, denoted as $\overline{g}(\underline{\mathbf{x}}, \overline{\mathbf{x}})$, such that

$$\overline{g}(\underline{\mathbf{x}}, \overline{\mathbf{x}}) \geq g(\mathbf{x}) \ (\forall \mathbf{x}, \underline{\mathbf{x}} \leq \mathbf{x} \leq \overline{\mathbf{x}}). \tag{5}$$

Thereby, $\overline{g}(\underline{\mathbf{x}}, \overline{\mathbf{x}})$ is a verifiable upper bound on the worst-case violation of Eq. (1) for data points in $[\underline{\mathbf{x}}, \overline{\mathbf{x}}]$. To compute $\overline{g}(\underline{\mathbf{x}}, \overline{\mathbf{x}})$, we mainly use the CROWN (Zhang et al., 2018; 2020) algorithm which is based on linear relaxation-based bound propagation as mentioned in Section 2, while we also use a more simple Interval Bound Propagation (IBP) (Gowal et al., 2018; Mirman et al., 2018) algorithm to compute the intermediate bounds of the hidden layers in NNs. Such intermediate bounds are required by CROWN to derive linear relaxation for nonlinear operators including activation functions, as well as nonlinear computation in the dynamics of the dynamical system. We use IBP on hidden layers for more efficient training and potentially easier optimization (Lee et al., 2021; Jovanović et al., 2021). Verified bounds computed in this way are differentiable, and then we aim to achieve $\overline{g}(\underline{\mathbf{x}}, \overline{\mathbf{x}}) \leq 0$ and minimize $\overline{g}(\underline{\mathbf{x}}, \overline{\mathbf{x}})$ in the training.

We additionally include a training objective term where we try to empirically find the worst-case violation of Eq. (1) by adversarial attack using projected gradient descent (PGD) (Madry et al., 2018), denoted as $\overline{g}^A(\underline{\mathbf{x}}, \overline{\mathbf{x}}) := g(A(\underline{\mathbf{x}}, \overline{\mathbf{x}}))$, where $A(\underline{\mathbf{x}}, \overline{\mathbf{x}}) \in \mathbb{R}^d$ $(\underline{\mathbf{x}} \leq A(\underline{\mathbf{x}}, \overline{\mathbf{x}}) \leq \overline{\mathbf{x}})$ is a data point found by PGD to empirically maximize $g(A(\underline{\mathbf{x}}, \overline{\mathbf{x}}))$ within the domain:

$$\underset{\mathbf{x} \in \mathbb{R}^d \ (\underline{\mathbf{x}} \leq \mathbf{x} \leq \overline{\mathbf{x}})}{\arg\max} g(\mathbf{x}), \tag{6}$$

but $A(\underline{\mathbf{x}}, \overline{\mathbf{x}})$ found by PGD is not guaranteed to be the optimal solution for Eq. (6). Since it is easier to train a model which empirically satisfies Eq. (1) compared to making Eq. (1) verifiable, we add this adversarial attack objective so that the training can more quickly reach a solution with most counterexamples eliminated, while certified training can focus on making it verifiable. This objective also helps to achieve that at least no counterexample can be empirically found, even if verified bounds by CROWN and IBP cannot yet verify all the examples in the current dataset $(\underline{\mathbf{x}}, \overline{\mathbf{x}}) \in \mathbb{D}$, as we may still be able to fully verify Eq. (1) at test time using a stronger verifier enhanced with large-scale branch-and-bound.

Overall, we optimize for a loss function to minimize the violation of $\overline{g}(\underline{\mathbf{x}}, \overline{\mathbf{x}})$ and $\overline{g}^A(\underline{\mathbf{x}}, \overline{\mathbf{x}})$:

$$L = \left( \mathbb{E}_{(\underline{\mathbf{x}}, \overline{\mathbf{x}}) \in \mathbb{D}} \ \sigma(\overline{g}(\underline{\mathbf{x}}, \overline{\mathbf{x}}) + \epsilon) + \lambda \max \sigma(\overline{g}^A(\underline{\mathbf{x}}, \overline{\mathbf{x}}) + \epsilon) \right) + L_{\text{extra}}, \tag{7}$$

where $\sigma(\cdot)$ is ReLU, $\epsilon$ is small value for ideally achieving Eq. (1) with a margin, as $\overline{g}(\underline{\mathbf{x}}, \overline{\mathbf{x}}) \leq -\epsilon$ and $\overline{g}^A(\underline{\mathbf{x}}, \overline{\mathbf{x}}) \leq -\epsilon$, $\lambda$ is a coefficient used to for assigning a weight to the PGD term, and $L_{\text{extra}}$ is an extra loss term which can be used to control additional properties of the model. After the training, the desired properties as Eq. (1) are verified by a formal verifier such as $\alpha,\beta$-CROWN with larger-scale branch-and-bound, and thus the soundness of the trained models can be guaranteed as long as the verification succeeds at test time.

We have formulated our general training framework in this section, and we will instantiate our training framework on the particular task of learning Lyapunov-stable neural controllers in Section 3.4.

### 3.3 TRAINING-TIME BRANCH-AND-BOUND

We now discuss how we initialize the training dataset $\mathbb{D}$ and dynamically maintain the dataset during the training by splitting hard examples into smaller subregions.

**Initial splits.** We initialize $\mathbb{D}$ by splitting the original input region-of-interest $\mathcal{B}$ into grids along each of its $d$ dimensions, respectively. We control the maximum size of the initial regions with a threshold $l$ which denotes the maximum length of each input dimension. For each input dimension $i$ ($1 \leq i \leq d$), we uniformly split the input range $[\underline{\mathbf{b}}_i, \overline{\mathbf{b}}_i]$ into $m_i = \lceil \frac{\overline{\mathbf{b}}_i - \underline{\mathbf{b}}_i}{l} \rceil$ segments in the initial split, such that the length of each segment is no larger than the threshold $l$. We thereby create $\prod_{i=1}^{d} m_i$ regions to initialize $\mathbb{D}$, where each region is created by taking a segment from each input dimension, respectively. Each region $(\underline{\mathbf{x}}, \overline{\mathbf{x}}) \in \mathbb{D}$ is also an example in the training dataset. We set the threshold $l$ such that the initial examples fill 1$\sim$2 batches according to the batch size, so that the batch size can remain stable in the beginning of the training rather than start with a small actual batch size.

**Splits during the training.** After we create the initial splits with uniform splits along each input dimension, during the training, we also dynamically split hard regions into even smaller subregions. We take dynamic splits instead of simply taking more initial splits, as we can leverage the useful information during the training to identify hard examples to split where the specification has not been verified, and we also identify the input dimension to split such that it can lead to the best improvement on the loss values.

In each training batch, we take each example $(\underline{\mathbf{x}}^{(k)}, \overline{\mathbf{x}}^{(k)})$ with $\overline{g}(\underline{\mathbf{x}}^{(k)}, \overline{\mathbf{x}}^{(k)}) > 0$, i.e., we have not been able to verify that $g(\mathbf{x}) \leq 0$ within the region $[\underline{\mathbf{x}}^{(k)}, \overline{\mathbf{x}}^{(k)}]$. We then choose one of the input dimensions $i(1 \leq i \leq d)$ and uniformly split the region into two subregions along the chosen input dimension $i$. At dimension $i$, suppose the original input range for the example is $[\underline{\mathbf{x}}_i^{(k)}, \overline{\mathbf{x}}_i^{(k)}]$, we split it into $[\underline{\mathbf{x}}_i^{(k)}, \frac{\underline{\mathbf{x}}_i^{(k)} + \overline{\mathbf{x}}_i^{(k)}}{2}]$ and $[\frac{\underline{\mathbf{x}}_i^{(k)} + \overline{\mathbf{x}}_i^{(k)}}{2}, \overline{\mathbf{x}}_i^{(k)}]$, while leaving other input dimensions unchanged. We remove the original example from the dataset and add the two new subregions into the dataset.

In order to maximize the benefit of splitting an example, we decide the input dimension to choose by trying each of the input dimensions $j(1 \leq j \leq d)$ and computing the total loss of the two new subregions when dimension $j$ is split. Suppose $L(\underline{\mathbf{x}}^{(k)}, \overline{\mathbf{x}}^{(k)})$ is the contribution of an example $(\underline{\mathbf{x}}^{(k)}, \overline{\mathbf{x}}^{(k)})$ to the loss function in Eq. (7). We take the dimension $j$ to split which leads to the lowest loss value for the new examples:

$$\underset{1 \leq j \leq d}{\arg\min} \ L(\underline{\mathbf{x}}^{(k)}, \overline{\mathbf{x}}^{(k,j)}) + L(\underline{\mathbf{x}}^{(k,j)}, \overline{\mathbf{x}}^{(k)}), \quad \text{where } \underline{\mathbf{x}}_j^{(k,j)} = \overline{\mathbf{x}}_j^{(k,j)} = \frac{\underline{\mathbf{x}}_j^{(k)} + \overline{\mathbf{x}}_j^{(k)}}{2}, \tag{8}$$

and $\underline{\mathbf{x}}_i^{(k,j)} = \underline{\mathbf{x}}_i^{(k)}, \overline{\mathbf{x}}_i^{(k,j)} = \overline{\mathbf{x}}_i^{(k)}$ keep unchanged for other dimensions $i \neq j$ not being split. All the examples requiring a split in a batch and all the input dimensions to consider for the split can be handled in parallel on GPU.

### 3.4 Modeling and Training Objectives for Lyapunov-stable Neural Control

To demonstrate our new certified training framework, we focus on its application on learning verifiably Lyapunov-stable neural controllers with state feedback. Since our focus is on a new certified training framework, we use the same model architecture as Yang et al. (2024). We use a fully-connected NN for the controller $u(\mathbf{x})$; for the Lyapunov function $V(\mathbf{x})$, we either use a model based on a fully-connected NN $\phi(\mathbf{x})$ as $V(\mathbf{x}) = |\phi(\mathbf{x}) - \phi(\mathbf{x}^*)| + \|(\epsilon_V I + R^\top R)(\mathbf{x} - \mathbf{x}^*)\|_1$, or a quadratic function as $V(\mathbf{x}) = (\mathbf{x} - \mathbf{x}^*)^\top (\epsilon_V I + R^\top R)(\mathbf{x} - \mathbf{x}^*)$, where $R \in \mathbb{R}^{n_r \times n_r}$ is an optimizable matrix parameter, and $\epsilon_V > 0$ is a small positive value to guarantee that $\epsilon_V I + R^\top R$ is positive definite. The construction of the Lyapunov functions automatically guarantees that $V(\mathbf{x}^*) = 0$ and $V(\mathbf{x}) > 0$ $(\forall \mathbf{x} \neq \mathbf{x}^*)$ (Yang et al., 2024) required in the Lyapunov condition.

We have discussed the formulation of $g(\mathbf{x})$ in Eq. (4). When bounding the violation term $V(\mathbf{x}_{t+1}) - (1 - \kappa)V(\mathbf{x}_t)$ in Eq. (4), we additionally apply a constraint $V(\mathbf{x}_{t+1}) \geq \rho + \epsilon$ for $\mathbf{x}_{t+1} \notin \mathcal{B}$. It is to prevent wrongly minimizing the violation by going out of the region-of-interest as $\mathbf{x}_{t+1} \notin \mathcal{B}$ while making $V(\mathbf{x}_{t+1})$ $(\mathbf{x}_{t+1} \notin \mathcal{B})$ small, such that the violation $V(\mathbf{x}_{t+1}) - (1 - \kappa)V(\mathbf{x}_t)$ appears to be small yet missing the $\mathbf{x}_{t+1} \in \mathcal{B}$ requirement.

As mentioned in Eq. (4), an additional term $L_{\text{extra}}$ can be added to control additional properties of the model. We use the extra loss term to control the size of the region-of-attraction (ROA). We aim to have a good proportion of data points from the region-of-interest $\mathbf{x} \in \mathcal{B}$, such that their Lyapunov function values are within the sublevel set $V(\mathbf{x}) < \rho$ where the Lyapunov condition is to be guaranteed. To do this, we randomly draw a batch of $n_\rho$ samples within $\mathcal{B}$, as $\tilde{\mathbf{x}}_1, \tilde{\mathbf{x}}_2, \cdots, \tilde{\mathbf{x}}_{n_\rho} \in \mathcal{B}$, and we define $L_{\text{extra}}$ as:

$$L_{\text{extra}} = \mathbb{I}\left(\frac{1}{n_\rho} \sum_{i=1}^{n_\rho} \mathbb{I}(V(\tilde{\mathbf{x}}_i) < \rho) < \rho_{\text{ratio}}\right) \frac{\lambda_\rho}{n_\rho} \sum_{i=1}^{n_\rho} \sigma(V(\tilde{\mathbf{x}}_i) + \rho - \epsilon), \tag{9}$$

which penalizes samples with $V(\tilde{\mathbf{x}}_i) > \rho - \epsilon$ when the ratio of samples within the sublevel set is below the threshold $\rho_{\text{ratio}}$, where $\epsilon$ is a small value for the margin as similarly used in Eq. (7) and $\lambda_\rho$ is the weight of term $L_{\text{extra}}$ Eq. (7). In our implementation, we simply fix $\rho = 1$ and make $n_\rho$ equal to the batch size of the training. The threshold $\rho_{\text{ratio}}$ and the weight $\lambda_\rho$ can be set to reach the desired ROA size, but setting a stricter requirement on the ROA size can naturally increase the difficulty of training.

All of our models are randomly initialized and trained from scratch. This provides an additional benefit compared to previous works (Wu et al., 2023; Yang et al., 2024) which commonly required an initialization for a traditional non-learning method (linear quadratic regulartor, LQR) with a small ROA. Yang et al. (2024) also proposed to enlarge ROA with carefully selected candidates states which are desired to be within the ROA by referring to LQR solutions. In contrast, our training does not require any baseline solution. Thus, this improvement from our method can reduce the burden of applying our method without requiring a special initialization.

## 4 Experiments

### 4.1 Experimental Settings

**Dynamical systems.** We demonstrate our new certified training work on learning Lyapunov-stable neural controllers with state feedback in several nonlinear discrete-time dynamical systems following Wu et al. (2023); Yang et al. (2024), as listed in Table 1: *Inverted pendulum* is about swinging up the pendulum to the upright equilibrium; *Path tracking* is about tracking a path for a planar vehicle; and *2D quadrotor* is about hovering a quadrotor at the equilibrium state. For inverted pendulum and path tracking, there are two different limits on the maximum allowed torque of the controller, where the setting is more challenging with a smaller torque limit. Detailed definition of the system dynamics ($f$ in Eq. (2)) is available in existing works: Wu et al. (2023) for inverted pendulum and path tracking, and Tedrake (2009) for 2D quadrotor.

**Implementation.** We use the PyTorch library auto_LiRPA (Xu et al., 2020) to compute CROWN and IBP verified bounds during the training. After a model is trained, we use $\alpha,\beta$-CROWN to finally verify the trained model, where $\alpha,\beta$-CROWN is configured to use verified bounds by auto_LiRPA

Table 1: Dynamical systems used in the experiments. All these settings follow Yang et al. (2024). $d$ means the dimension of input states and $n_u$ means the dimension of control input which is from the output of the controller. There is a limit on the control input $u$ and the output of the controller is clamped according to the limit, where some symbols in the limit on $u$ are from the dynamics of the systems: $m$ for mass, $g$ for gravity, $l$ for length, and $v$ for velocity. Size of the region-of-interest here is represented by the upper boundary $\overline{\mathbf{b}}$, and $\underline{\mathbf{b}} = -\overline{\mathbf{b}}$ holds for all the systems here. Equilibrium state of all the systems here is $\mathbf{x}^* = \mathbf{0}$.

| System | $d$ | $n_u$ | Limit on $u$ | Region-of-interest |
|--------|-----|-------|--------------|--------------------|
| Inverted pendulum | 2 | 1 | $\|u\| \leq 8.15 \cdot mgl$ (large torque)
$\|u\| \leq 1.02 \cdot mgl$ (small torque) | $[12, 12]$ |
| Path tracking | 2 | 1 | $\|u\| \leq 1.68 \cdot l/v$ (large torque)
$\|u\| \leq l/v$ (small torque) | $[3, 3]$ |
| 2D quadrotor | 6 | 3 | $\|u\|_\infty \leq 1.25 \cdot mg$ | $[0.75, 0.75, \pi, 2, 4, 4, 3]$ |

Table 2: Comparison on the verification time cost and the size of ROA. "Pendulum" refers to the inverted pendulum system. Model checkpoints for Wu et al. (2023) are obtained from the source code of Yang et al. (2024) and the same models have been used for comparison in Yang et al. (2024), where "-" denotes that on some of the systems models for Wu et al. (2023) are not available. Yang et al. (2024) and ours have the same model architecture on each system.

| System | Wu et al. (2023) | | CEGIS (Yang et al., 2024) | | Ours | |
|--------|------|-----|------|-----|------|-----|
| | Time | ROA | Time | ROA | Time | ROA |
| Pendulum (large torque) | 11.3s | 53.28 | 33s | 239.04 | 32s | **495.36** |
| Pendulum (small torque) | - | - | 25s | 187.20 | 26s | **275.04** |
| Path tracking (large torque) | 11.7s | 14.38 | 39s | 18.27 | **31s** | **21.60** |
| Path tracking (small torque) | - | - | 34s | 10.53 | **27s** | **11.51** |
| 2D quadrotor | - | - | 1.1hrs | 3.29 | **11.5min** | **54.39** |

and run branch-and-bound on the input space to tighten the verified bounds until the verification succeeds, which has been used in the same way in Yang et al. (2024). Additional details of the experiments are included in Appendix A.

## 4.2 MAIN RESULTS

We show the main results in Table 2, where we compare the verification time cost and size of ROA with the previous state-of-the-art method based on CEGIS (Yang et al., 2024), as well as an earlier work (Wu et al., 2023) on applicable systems. Following Wu et al. (2023), we estimate the size of ROA by considering grid points in the region-of-interest $\mathcal{B}$ and counting the proportion of grid points within the sublevel set of the Lyapunov function where the Lyapunov condition is verified, multiplied by the volume of $\mathcal{B}$. Models by Wu et al. (2023) have much smaller ROA than Yang et al. (2024), and thus we focus on comparing our method with Yang et al. (2024). On inverted pendulum, our method produces much larger ROA with similar verification time, and on path tracking, our method produces larger ROA while also reducing the verification time. On these two systems, the verification time cannot be greatly reduced, due to the overhead of launching $\alpha,\beta$-CROWN and low GPU utilization when the verification is relatively easy. On 2D quadrotor with a much higher difficulty, our method significantly reduces the verification time (11.5 minutes compared to 1.1 hours by Yang et al. (2024)) while also significantly enlarging the ROA (54.39 compared to 3.29 by Yang et al. (2024)). These results demonstrate the effectiveness of our method on producing verification-friendly Lyapunov-stable neural controllers and Lyapunov functions with larger ROA. In Figure 1, we visualize the ROA on 2D quadrotor, with different 2D views, which demosntrates a larger ROA compared to the Yang et al. (2024) baseline. In Appendix B, we visualize the distribution of the subregions after our training-time branch-and-bound, which suggests that much more extensive splits tend to happen when at least one of the input states is close to that of the equilibrium state, where Lyapunov function values are relatively small and the training tends to be more challenging.

Table 3: Runtime of training, size of the training dataset, and the ratio of examples in the training dataset verifiable by CROWN without further branch-and-bound. "Initial dataset size" denotes the size of the training dataset at the start of the training, and "final dataset size" denote the size at the end of the training. All the models can be fully verified at test time using $\alpha,\beta$-CROWN with branch-and-bound at the input space, as shown in Table 2.

| System | Runtime | Initial dataset size | Final dataset size | Verified by CROWN |
|---|---|---|---|---|
| Pendulum (large torque) | 6min | 58080 | 68686 | 100% |
| Pendulum (small torque) | 32min | 58080 | 657043 | 100% |
| Path tracking (large torque) | 17min | 40400 | 7586381 | 94.95% |
| Path tracking (small torque) | 16min | 40400 | 222831 | 99.97% |
| 2D quadrotor | 107min | 46336 | 34092930 | 88.18% |

Table 4: Training and test results of ablation study conducted on the 2D quadrotor system. For training results, we report the dataset size at the end of the training and the ratio of training examples verified by CROWN, where "verified (all)" is evaluated on all the training examples, while "verified (within the sublevel set)" excludes examples verified to be out of the sublevel set with $V(\mathbf{x}) < \rho$. For test results, we report if the model can be fully verified at test time by $\alpha,\beta$-CROWN and a "candidate ROA" size which denotes the volume of the sublevel set with $V(\mathbf{x}) < \rho$. "Candidate ROA" is the true ROA if the model is fully verified.

| Method | Training | | | Test | |
|---|---|---|---|---|---|
| | Dataset size | Verified (all) | Verified (within the sublevel set) | Fully verified | Candidate ROA |
| Default | 34092930 | 88.18% | **86.95%** | **Yes** | **54.39** |
| No dynamic split | 64916160 | 99.95% | 38.29% | No | 0.08 |
| Naive dynamic split | 20477068 | 90.05% | 55.62% | No | 0.0095 |

In Table 3, we show information about the training, including the time cost of training, size of the dynamic training dataset and the ratio of training examples which can be verified using verified bounds by CROWN (Zhang et al., 2018; Xu et al., 2020) at the end of the training. Our training dataset is dynamically maintained and expanded as described in Section 3.3, and the dataset size grows from the "initial dataset size" to the "final dataset size" shown in Table 3. At the end of the training, most of the training examples (more than 88%) can already be verified by CROWN bounds. Although not all of the training examples are verifiable by CROWN, all the models can be fully verified when we use $\alpha,\beta$-CROWN to finally verify the models at test time, where $\alpha,\beta$-CROWN further conducts branch-and-bound on the input space using CROWN bounds.

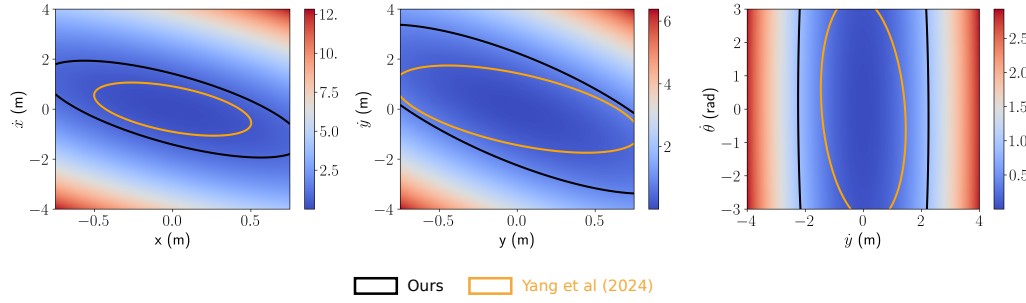

Figure 1: Visualization of the Lyapunov function (color plots) and ROA (contours) on the 2D quadrotor system with three different 2D views compared to Yang et al. (2024). The system contains 6 states denoted as $\mathbf{x} = [x, y, \theta, \dot{x}, \dot{y}, \dot{\theta}]$. Our method demonstrates a 16X larger ROA (in terms of the volume of ROA on the 6-dimensional input space) compared to Yang et al. (2024).

## 4.3 ABLATION STUDY

In this section, we conduct an ablation study to demonstrate the necessity of using our dynamic splits to maintain the training dataset as described in Section 3.3, on the largest 2D quadrotor system. We consider two variations of our proposed method: *No dynamic split* means that we use a large number of initial splits by reducing the threshold $l$ which controls the maximize size of initial regions mentioned in Section 3.3, and the dataset is then fixed and there is no dynamic split throughout the training; *Naive dynamic split* means that we use dynamic splits but we simply split along the input dimension with the largest size, as $\arg\max_{1 \le j \le d}(\overline{\mathbf{x}}_i^{(k)} - \underline{\mathbf{x}}_j^{(k)})$, instead of taking the best input dimension in terms of reducing the loss value as Eq. (8). We show the results in Table 4. Neither of "no dynamic split" and "naive dynamic split" can produce verifiable models. We observe that they both tend to make the sublevel set with $V(\mathbf{x}) < \rho$ very small, which leads to a very small ROA size even if the model can be verified (if the weight on the extra loss term for ROA in Eq. (9) is increased, the training does not converge with many counterexamples which can be empirically found). For the two variations, although most of the training examples can still be verified at the end of training, if we check nontrivial examples which are not verified to be out of the sublevel set (see "verified (within the sublevel set)" in Table 4), a much smaller proportion of these examples are verified. Without our proposed dynamic splits decided by Eq. (8), these two variations cannot identify hard examples to split and split along the best input dimension to efficiently ease the training, leaving many unverified examples among those possibly within the sublevel set, despite that the size of the sublevel set is significantly shrunk. This experiment demonstrates the benefit of our proposed dynamic splits.

## 5 CONCLUSION

To conclude, we propose a new certified training framework for training verification-friendly models where a relatively global guarantee can be verified for an entire region-of-interest in the input space. We maintain a dynamic dataset of subregions which cover the region-of-interest, and we split hard examples into smaller subregions throughout the training, to ease the training with tighter verified bounds. We demonstrate our new certified training framework on the problem of learning and verifying Lyapunov-stable neural controllers. We show that our new method produces more verification-friendly models which can be more efficiently verified at test time while the region-of-attraction also becomes much larger compared to the state-of-the-art baseline.

A limitation of this work is that only low-dimensional dynamical systems have been considered, which is also a common limitation of previous works on this Lyapunov problem (Chang et al., 2019; Wu et al., 2023; Yang et al., 2024). Future works may consider scaling up our method to higher-dimensional systems. Since splitting regions on the input space can become less efficient if the dimension of the input space significantly increases, future works may consider applying splits on the intermediate bounds of activation functions (potentially with sparsity), which has been commonly used in state-of-the-art NN verifiers (mentioned in Section 2) for verifying trained models on high-dimensional tasks such as image classification.

Although our new certified training framework is generally formulated, we have only focused on demonstrating the training framework on Lyapunov asymptotic stability. Given the generality of our new framework, it has the potential to enable broader applications, such as other safety properties including reachability and forward invariance mentioned in Section 2, control systems with more complicated settings such as output feedback systems, or even applications beyond control. These will be interesting directions for future work.

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

# A  DETAILS OF THE IMPLEMENTATION AND EXPERIMENTS

We directly adopt the model architecture of all the controllers and Lyapunov functions from Yang et al. (2024) (we follow their source code which has some minor difference with the information provided in their paper). The controller is always a fully-connected NN with 8 hidden neurons in each hidden layer. For inverted pendulum and path tracking, there are 4 layers, and for 2D quadrotor, there are 2 layers. ReLU is used as the activation function. A NN-based Lyapunov function is used for inverted pendulum and path tracking, where the NN is a fully-connected NN with 4 layers, and the number of hidden neurons is 16, 16, and 8 for the three hidden layers, respectively. Leaky ReLU is used as the activation function for NN-based Lyapunov functions. A quadratic Lyapunov function with $n_r = 6$ is used for 2D quadrotor. For $\kappa$ in Eq. (3), $\kappa = 0.001$ is used for inverted pendulum and path tracking, and $\kappa = 0$ is used for 2D quadrotor, following Yang et al. (2024).

We use a batch size of 30000 for all the training. We mainly use a learning rate of $5 \times 10^{-3}$, except $2 \times 10^{-2}$ for path tracking. In the loss function, we set $\lambda$ to $10^{-4}$, $\lambda_p$ to 0.1, and $\epsilon$ to 0.01. We try to make $\rho_{\text{ratio}}$ as large as possible for individual systems, as long as the training works. We set $\rho_{\text{ratio}} = 0.1$ for 2D quadrotor. For inverted pendulum and path tracking, the range of $\rho_{\text{ratio}}$ is between 0.5 and 0.9 for different settings. We start our dynamic splits after 100 initial training steps and continue until 5000 training steps (for 2D quadrotor) or if the training finishes before that (for other systems). For the adversarial attack, we use PGD with 10 steps and a step size of 0.25 relative to the size of subregion. We fix $\rho = 1.0$ during the training. At test time, we slightly reduce $\rho$ to 0.9 for 2D quadrotor while we keep $\rho = 1.0$ for other systems. Using a slightly smaller $\rho$ at test time instead of the value used for training has been similarly done in Yang et al. (2024) to ease the verification. Each training is done using a single NVIDIA GeForce RTX 2080 Ti GPU, while the verification with $\alpha,\beta$-CROWN at test time is done on a NVIDIA RTX A6000 GPU which is the same GPU model used by Yang et al. (2024).

# B  VISUALIZATION OF BRANCH-AND-BOUND

In this section, we visualize the distribution of subregions in the training dataset $\mathbb{D}$ at the end of the training, in order to understand where the most extensive branch-and-bound happens. Specifically, we check the distribution of the center of subregions. For systems with two input states (inverted pendulum and path tracking), we use 2D histogram plots, as shown in Figure 2 and 3. For the 2D quadrotor system which has 6 input states (and thus a 2D histogram plot cannot be directly used), we plot the distribution for different measurements of the subregion centers, including the $\ell_1$ norm, $\ell_\infty$ norm, and the minimum magnitude over all the dimensions, as shown in Figure 4. We find that much more extensive splits tend to happen when at least one of the input states is close to that of the equilibrium state. Such areas have relatively small Lyapunov function values and tend to be more challenging for the training and verification. Specifically, in Figure 2a, 3a and 3b, extensive splits happen right close to the equilibrium state, while in Figure 2b, although extensive splits are not fully near the equilibrium state, extensive splits happen for subregions where the value for the $\dot{\theta}$ input state is close to 0 (i.e., value of $\dot{\theta}$ for the equilibrium state). The observation is also similar for the 2D quadrotor system, where Figure 4c shows that most subregions have at least one input state close to 0.

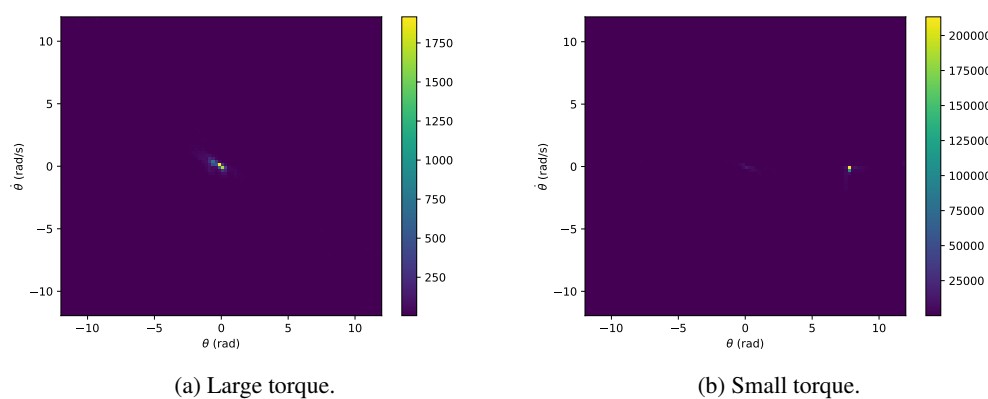

(a) Large torque.

(b) Small torque.

Figure 2: Visualization for the distribution of subregions in $\mathbb{D}$ at the end of the training for the inverted pendulum system, with large torque limit and small torque limit, respectively. The 2D histogram plots show the distribution of the center of subregions. $\theta$ and $\dot{\theta}$ denote the angular position and angular velocity, respectively, for the two input states in inverted pendulum.

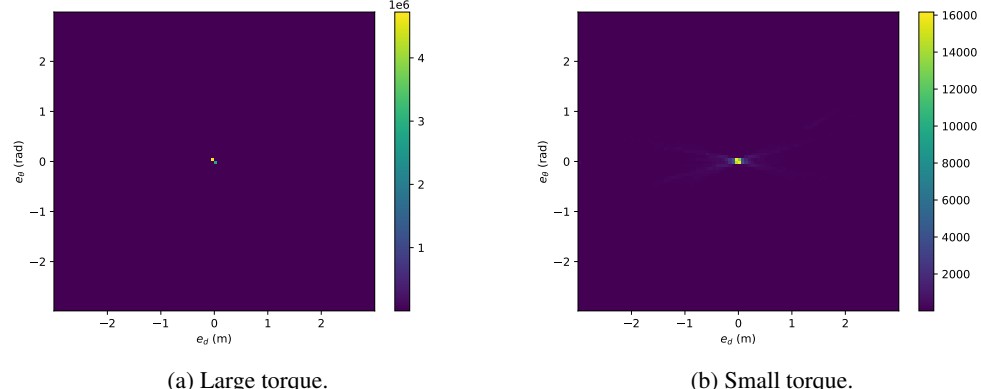

(a) Large torque.

(b) Small torque.

Figure 3: Visualization for the distribution of subregions in $\mathbb{D}$ at the end of the training for the path tracking system, similar to Figure 2. $e_d$ and $e_\theta$ denote the distance error and angle error, respectively, for the two input states in path tracking.

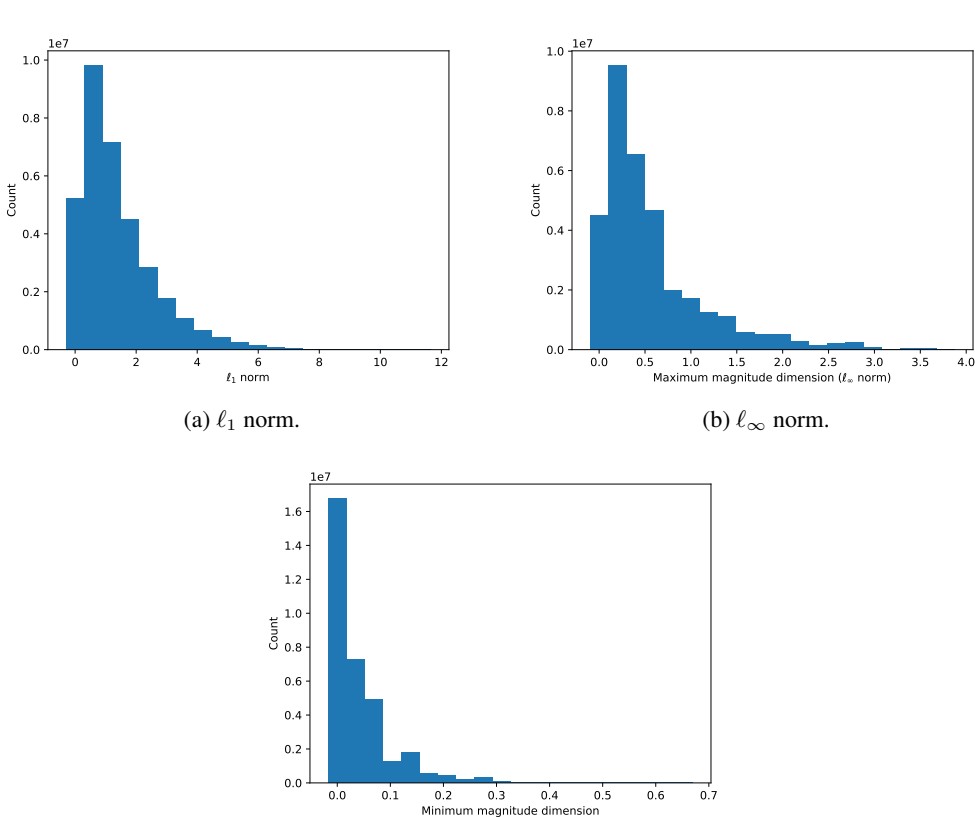

(a) $\ell_1$ norm.

(b) $\ell_\infty$ norm.

(c) Minimum magnitude over all the dimensions.

Figure 4: Visualization for the distribution of subregions in $\mathbb{D}$ at the end of the training for the 2D quadrotor system. We check the distribution of $\ell_1$ norm, $\ell_\infty$ norm, and the minimum magnitude over all the dimensions (all the input states), respectively, for the subregion centers.

