# OpenReview forum: "Certified Training with Branch-and-Bound: A Case Study on Lyapunov-stable Neural Control"
_ICLR.cc/2025/Conference — ICLR 2025 Conference Withdrawn Submission_

### Official Review · Reviewer_efAm · 2024-10-19

**Soundness:** 2
**Presentation:** 1
**Contribution:** 2
**Rating:** 5
**Confidence:** 4

**Summary:**

This paper aims to produce Lyapunov-stable neural controllers. The novel aspect of the work is using a branch-and-bound approach during training to generate samples. The authors evaluate their method using case studies on several dynamical systems, demonstrating a significant reduction in verification time and an expansion of the region of attraction (ROA) compared to existing methods.

**Strengths:**

The experiment result shows the proposed approach outperforms the previous work.

**Weaknesses:**

This work has the following major issues.

1. The background and related work introduction is not proper.
    - 1.1. The discussion of a whole branch of work on barrier function/control barrier function-based controller training is largely missing or in a misleading way. Barrier function and control barrier function techniques for safety are very similar to Lyapunov function techniques for stability in terms of their formulation in control theory. There have been extensive works on neural barrier function learning or barrier/policy joint learning [1,2,3]. The authors misinterpret these works by claiming that they, including [2], "did not provide formal guarantees". In fact, most of these works [1,2] explicitly mention they have an additional verification step to ensure the correctness of their approaches (Page 10 in [1] and Page 6 in [2]). It raises concerns about the credibility of this work. Please properly discuss these works.
    - 1.2. The local/global robustness verification is not quite relevant to this work, and the discussion should be eliminated from the related work. Instead, the verification of neural-network controlled systems should be discussed, as they are in the same track of this work, e.g., [4,5,6].

2. The branch-and-bound section is not very clear.
    - 2.1. Is the training dataset $\mathbb{D}$ a set of points or regions? What does it mean by $(\underline{x},\underline{x})$ in Line 269?
    - 2.2. Does the proposed approach need to verify that $g(x)\leq 0$ for every region? If so, then how does this work scale to high-dimensional systems?

3. Empirical comparison with STOAs is expected.
    - 3.1. ROA considered in this work is also very relevant to the invariant set in control theory. Thus, this work is expected to compare with STOAs in this domain, e.g., [7].

[1] Zhao, Hengjun, Xia Zeng, Taolue Chen, Zhiming Liu, and Jim Woodcock. "Learning Safe Neural Network Controllers with Barrier Certificates." arXiv preprint arXiv:2009.09826 (2020).

[2] Jin, Wanxin, Zhaoran Wang, Zhuoran Yang, and Shaoshuai Mou. "Neural certificates for safe control policies." arXiv preprint arXiv:2006.08465 (2020).

[3] Wang, Yixuan, Simon Sinong Zhan, Ruochen Jiao, Zhilu Wang, Wanxin Jin, Zhuoran Yang, Zhaoran Wang, Chao Huang, and Qi Zhu. "Enforcing Hard Constraints with Soft Barriers: Safe Reinforcement Learning in Unknown Stochastic Environments." arXiv preprint arXiv:2209.15090 (2022).

[4] Ivanov, Radoslav, Taylor Carpenter, James Weimer, Rajeev Alur, George Pappas, and Insup Lee. "Verisig 2.0: Verification of neural network controllers using taylor model preconditioning." In International Conference on Computer Aided Verification, pp. 249-262. Cham: Springer International Publishing, 2021.

[5] Huang, Chao, Jiameng Fan, Xin Chen, Wenchao Li, and Qi Zhu. "Polar: A polynomial arithmetic framework for verifying neural-network controlled systems." In International Symposium on Automated Technology for Verification and Analysis, pp. 414-430. Cham: Springer International Publishing, 2022.

[6] Teuber, Samuel, Stefan Mitsch, and André Platzer. "Provably Safe Neural Network Controllers via Differential Dynamic Logic." arXiv preprint arXiv:2402.10998 (2024).

[7] Harapanahalli, Akash, and Samuel Coogan. "Certified Robust Invariant Polytope Training in Neural Controlled ODEs." arXiv preprint arXiv:2408.01273 (2024).

**Questions:**

Please see the weakness.

---

> ### Author Response · Authors · 2024-11-16
> **Rebuttal (1/3): summary of our rebuttal; why Jin et al., 2020 in fact did not provide formal guarantees**
>
> We appreciate your constructive feedback and the detailed list of references. We wanted to provide an early response since we believe most weaknesses mentioned are due to misunderstandings and misconceptions. We have clarified why some previous work (Jin et al., 2020\) did not actually achieve formal guarantees. We also clarified the difference between barrier functions and Lyapunov functions, and we have extended the discussion of the related work of our paper (see the updated PDF), cited the papers you mentioned, and discussed the key differences. We believe that none of these papers on barrier functions are valid baselines for empirical comparison, since our setting on Lyapunov stability is clearly different. We also provided a clarification on branch-and-bound.
>
> ## Some previous works without formal verification
>
> We thank the reviewer for raising this point for us to make our related work section more clear and comprehensive. But we would like to clarify that we believe it is the fact that several works we cited (Jin et al., 2020; Sun & Wu, 2021; Dawson et al., 2022; Liu et al., 2023), especially Jin et al., 2020 which the reviewer has mentioned, **did not achieve formal guarantees**, and our writing is **not** “in a misleading way”. It is clear that Sun & Wu, 2021; Dawson et al., 2022; Liu et al., 2023 did not consider formal verification. We will explain Jin et al., 2020 in more detail.
>
> >There have been extensive works on neural barrier function learning or barrier/policy joint learning \[1,2,3\]. The authors misinterpret these works by claiming that they, including \[2\], "did not provide formal guarantees".
>
> **We believe we did not misinterpret them.** We did not say \[1, 3\] which are not about Lyapunov stability “did not provide formal guarantees”. Please refer to our response in the [“Related works on other safety guarantees in control” section](https://openreview.net/forum?id=8ctju6iFcn&noteId=07uUCvZMZu) regarding related works on barrier functions which are different from Lyapunov functions for stability.
>
> Now we will explain why **Jin et al., 2020 (\[2\] in the review) did not achieve formal guarantees.** The lack of formal guarantees in Jin et al., 2020 **has also been confirmed by multiple previous works**: Edwards et al., 2023; Abate et al., 2024; Yang et al., 2024 have all mentioned that Jin et al., 2020 is either unsound (regarding the certificates) or did not provide formal guarantees.
>
> The reviewer has mentioned page 6 in Jin et al., 2020 which appeared to discuss verification. However, while Jin et al., 2020 has theoretically aimed to achieve a verification and included a discussion on the verification, their theoretical explanation is based on an *assumption* that the model is Lipschitz continuous and a (sound) Lipschitz constant is available. **In their actual implementation, they only empirically checked a finite number of samples, without actually computing the Lipschitz constant** which their verification scheme depends on. Thus, they have not achieved a formal verification, especially computing sound and sufficiently tight Lipschitz constants is nontrivial and could be challenging in practice (Jordan et al., 2020; Shi et al., 2022). Specifically, in Page 6 in Jin et al., 2020, it is mentioned that “we only verify if $ \\min\_{i=1,2,\\cdots} V\_{\\boldsymbol{\\omega}}(\\mathbf{x}^i\_{g})\\leq \\epsilon\_2$, where $\\epsilon\_2\>0$ is a small constant parameter that bounds the tolerable numerical error, and $\\mathbf{x}^i\_g$ are the points in the discretization of $\\mathcal{\\bar{X}}\_g$”. We have revised the paper to explain it.
>
> Jin, Wanxin, Zhaoran Wang, Zhuoran Yang, and Shaoshuai Mou. "Neural certificates for safe control policies." arXiv preprint arXiv:2006.08465 (2020).
>
> Edwards, A., Peruffo, A., & Abate, A. (2023). A General Framework for Verification and Control of Dynamical Models via Certificate Synthesis. arxiv: 2309.06090
>
> Abate, A., Bogomolov, S., Edwards, A., Potomkin, K., Soudjani, S., & Zuliani, P. (2024). Safe Reach Set Computation via Neural Barrier Certificates. arXiv preprint arXiv:2404.18813.
>
> Yang, L., Dai, H., Shi, Z., Hsieh, C. J., Tedrake, R., & Zhang, H. Lyapunov-stable Neural Control for State and Output Feedback: A Novel Formulation. In Forty-first International Conference on Machine Learning.
>
> Jordan, M., & Dimakis, A. G. (2020). Exactly computing the local lipschitz constant of relu networks. Advances in Neural Information Processing Systems, 33, 7344-7353.
>
> Shi, Z., Wang, Y., Zhang, H., Kolter, J. Z., & Hsieh, C. J. (2022). Efficiently computing local lipschitz constants of neural networks via bound propagation. Advances in Neural Information Processing Systems, 35, 2350-2364.

---

> > ### Comment · Reviewer_efAm · 2024-11-19
> >
> > Thanks for the effort in the response. A few concerns are still remaining.
> >
> > 1. Thanks for emphasizing **Lyapunov** in the response and it reminded me of rechecking the definition. To my knowledge, **Lyapunov stability** denotes that the system will stay close to the equilibrium point, rather than converge to the equilibrium point, while **asymptotical stability** requires convergence. So is this work for asymptotic stability actually? Please correct me if I am wrong.
> >
> > 2. The authors claim that **Lyapunov functions and control barrier functions are different**, which I agree with. But it is also well acknowledged that these two techniques are different in encoding but very similar in computation, and are always discussed together [1,2]. Since this work does not propose new encoding techniques, it is expected that the authors have a comprehensive comparison with the works in both Lyapunov functions and control barrier functions, or clearly clarify the technical challenge of considering Lyapunov functions compared to control barrier functions. It helps identify the novelty of this work.
> >
> > [1] Anand, Akhil, Katrine Seel, Vilde Gjærum, Anne Håkansson, Haakon Robinson, and Aya Saad. "Safe learning for control using control Lyapunov functions and control barrier functions: A review." Procedia Computer Science 192 (2021): 3987-3997.
> >
> > [2] Romdlony, Muhammad Zakiyullah, and Bayu Jayawardhana. "Stabilization with guaranteed safety using control Lyapunov–barrier function." Automatica 66 (2016): 39-47.
> >
> > 3. I still think the writing of this paper is misleading. The paper is largely about safe reinforcement learning considering Lyapunov/Lyapunov-like functions. However, starting from the introduction, the authors try very hard to link their work with NN robustness, using NN robustness as the main background (Para. 2 in the introduction). Later in the main technical section -- Section 3.1, the authors again only emphasized NN robustness in the statement "Unlike previous certified training works, ..." to show the difference of their work with existing works. To conclude, the way of linking this work particularly with NN robustness is confusing to me. It brings the risk of making readers who are not control experts overly estimate the contribution of this work. It is more reasonable for me, as mentioned earlier to focus on safe RL and re-organize the story line.

---

> > > ### Author Response · Authors · 2024-11-20
> > > **Clarifications to address remaining concerns (1/2)**
> > >
> > > We thank you for the timely response and valuable feedback. We address your remaining concerns below.
> > >
> > > ## Asymptotic stability
> > >
> > > >To my knowledge, Lyapunov stability denotes that the system will stay close to the equilibrium point, rather than converge to the equilibrium point, while asymptotical stability requires convergence. So is this work for asymptotic stability actually?
> > >
> > > Thanks for clarifying the difference between these two stability notions, and you are right that this work is for Lyapunov *asymptotic* stability. We have revised our paper and uploaded a new PDF. We have clarified that we consider “Lyapunov asymptotic stability” and “asymptotic stability guarantees”. We previously followed prior works (Dai et al., 2021, Wu et al., 2023, Yang et al., 2024\) which also considered Lyapunov asymptotic stability but simply referred to it as “Lyapunov stability” without specifically mentioning “asymptotic”. We agree that it should be clarified that asymptotic stability is considered.
> > >
> > > ## Lyapunov functions v.s. control barrier functions
> > >
> > > >These two techniques are different in encoding but very similar in computation, and are always discussed together \[1,2\]. Since this work does not propose new encoding techniques, it is expected that the authors have a comprehensive comparison with the works in both Lyapunov functions and control barrier functions
> > >
> > > We understand that these two techniques are relevant and some works may discuss them together, although not “always”. Following your suggestions in your initial review, we have also revised our Section 2 to discuss related works on both two techniques. However, we believe that the necessity of more comprehensively comparing Lyapunov functions and barrier functions depends on the focus of a work. Our work is on a new training method, not the encoding of the control problem which we directly follow settings in Yang et al., 2024\. Since multiple previous works on training Lyapunov stable neural controllers (Dai et al., 2021, Wu et al., 2023, Yang et al., 2024\) also all focused on Lyapunov functions for stability, we believe it is also reasonable for us to focus on Lyapunov asymptotic stability for demonstrating our new training method.
> > >
> > > >clearly clarify the technical challenge of considering Lyapunov functions compared to control barrier functions. It helps identify the novelty of this work.
> > >
> > > The main challenge of considering Lyapunov functions is that Lyapunov asymptotic stability is a stronger guarantee, as we have clarified in our last reply, which makes the training and verification more challenging. However, the novelty of our work is not tied to Lyapunov, as our novelty is actually in proposing a new and novel training method which is generally formulated and enhanced with a novel training-time branch-and-bound (see our further discussions in the next section below). We use Lyapunov asymptotic stability as a case study to demonstrate the use of our new training framework.

---

> > > > ### Author Response · Authors · 2024-11-20
> > > > **Clarifications to address remaining concerns (2/2)**
> > > >
> > > > ## Storyline
> > > >
> > > > We thank you for providing perspectives as a control researcher. Although we agree that it is possible to re-organize the storyline by focusing on safe control, we believe that both our original storyline and the alternative one you suggested are viable, and we do not think our current paper is “misleading”.
> > > >
> > > > We want to clarify that our writing is actually consistent with what our title “Certified Training with Branch-and-Bound: A Case Study on Lyapunov-stable Neural Control” suggests. First, we propose a new training framework with a novel training-time branch-and-bound. Note that our training framework is generally formulated (Section 3.2 and Section 3.3) and the framework itself is not simply tied to Lyapunov asymptotic stability. Second, we demonstrate the use of our new training framework with a case study on Lyapunov asymptotic stability, as instantiated in Section 3.4, which is also our focus for the experiments.
> > > >
> > > > Although our experiments focus on Lyapunov asymptotic stability, our methodology itself is general and has the potential for broader impacts such that readers from other areas (“who are not control experts” as you mentioned) may consider applying our certified training framework for training verifiable models in various mission-critical applications other than safe control. Thus, we chose the current storyline, and we hope this can clarify our logic behind our storyline.
> > > >
> > > > >However, starting from the introduction, the authors try very hard to link their work with NN robustness, using NN robustness as the main background (Para. 2 in the introduction). Later in the main technical section \-- Section 3.1, the authors again only emphasized NN robustness in the statement "Unlike previous certified training works, ..." to show the difference of their work with existing works.
> > > >
> > > > Our discussion about general NN verification and certified training in the beginning of Section 2 and Section 3 is actually relevant to the Lyapunov asymptotic stability problem.  We think there might be some misunderstanding regarding the role of “NN robustness” in the paper. Note that NN verification is not simply about “NN robustness” which is a special case of NN verification which has broad applications including safe control. When we mention “robustness”, we are referring to the specific application of NN verification commonly considered in previous certified training works. However, our discussion there is focused on general NN verification and certified training, not just NN robustness. Such discussion is relevant to Lyapunov asymptotic stability, as using certified training for Lyapunov asymptotic stability is also a special case of our general training framework.

---

> > > > > ### Comment · Reviewer_efAm · 2024-11-20
> > > > >
> > > > > Thanks for the response. I raised the score as part of my concerns are addressed. But I am not convinced by the authors' further clarification on the writing and the concern about misleading thus still remains -- it is a safe RL paper for sure, but mainly using NN robustness to motivate the story and indicate the difference, while there is in fact a large amount of existing work on safe RL that the authors can and should build the story on.

---

> > > > > > ### Author Response · Authors · 2024-11-24
> > > > > > **Adjusted storyline**
> > > > > >
> > > > > > Thanks for raising the score, and again for your valuable feedback from the perspective of safe control. We have adjusted the storyline to motivate our work around safe control (please mainly see Section 1 and 2 in the updated PDF; due to the overall adjustment, we didn’t change the color of text).
> > > > > >
> > > > > > We now begin our introduction with safe control and its different desired properties (including reachability, forward invariance, and Lyapunov stability). Then we focus on the Lyapunov asymptotic stability and introduce its background and implications. Next, we discuss prior works on the same problem and their limitations, to motivate our introduction of certified training, where we also mention our difference compared to existing certified training. After that, we provide an overview of our method and our contributions, which remains the same as our previous version. We have also adjusted Section 2, to introduce related works on control (including both Lyapunov asymptotic stability and other safety properties) first.
> > > > > >
> > > > > > We hope you could check our revision and reconsider your rating.

---

> > > > > > > ### Author Response · Authors · 2024-11-30
> > > > > > > **Reminder to check our revision**
> > > > > > >
> > > > > > > As we are approaching the end of the discussion period, we would like to remind Reviewer efAm to check our latest revision and response and kindly consider updating the rating as suitable.
> > > > > > >
> > > > > > > As explained in our last reply, we have already adjusted the storyline in our revision as suggested by the reviewer. At this point, we believe we have addressed all the concerns raised by the reviewer.

---

> ### Author Response · Authors · 2024-11-16
> **Rebuttal (2/3): Lyapunov stability (with Lyapunov functions) and forward invariance (with barrier functions) are different guarantees**
>
> ## Related works on other safety guarantees in control
>
> We thank the reviewer for suggesting a more comprehensive related work section. We acknowledge that we missed some previous works on control barrier functions, as we mostly focused on related works on Lyapunov stability which is the main focus of our entire paper.
>
> However, we would like to clarify that **Lyapunov functions and control barrier functions are different, and Lyapunov stability is a stronger guarantee than forward invariance guaranteed by control barrier functions**. Lyapunov stability guarantees a convergence towards the equilibrium point, while forward invariance only guarantees that the system remains in the invariance set (without reaching an unsafe set) but it does not guarantee a convergence. Similar to previous works on the Lyapunov stability of neural controllers (such as Yang et al., 2024; Wu et al., 2023), we also mainly focused on the Lyapunov stability.
>
> We do agree that control barrier functions are still relevant as they also aim for the safety of controllers. We have added a new paragraph (highlighted in blue in the updated PDF file) in our related work section to discuss related works on other (i.e., non-Lyapunov) safety properties of neural controllers, and we have cited all the new references suggested by the reviewer. We agree that \[1\] cited in the review does contain formal verification by a SMT solver for control barrier functions (not Lyapunov functions).
>
> >The local/global robustness verification is not quite relevant to this work, and the discussion should be eliminated from the related work.
>
> We mentioned local/global robustness, as there are a large body of works on certified training which was originally proposed for local robustness. Since we consider certified training for neural controllers in this paper, we believe it is necessary to discuss the background of certified training, as well as our motivation of introducing training-time branch-and-bound for certified training, due to the significantly different problem for control here in contrast to robustness.
>
> ## ROA comparison
>
> > ROA considered in this work is also very relevant to the invariant set in control theory. Thus, this work is expected to compare with STOAs in this domain, e.g., \[7\].
>
> As we have clarified above, **Lyapunov stability and forward invariance are different and Lyapunov stability is a stronger guarantee, and thus we believe results on these two different types of safety guarantees are not comparable**.
>
> While a comparison with state-of-the-art is necessary, the scope should be restricted to works on Lyapunov stability. The existing state-of-the-art on learning Lyapunov-stable neural controllers is Yang et al., 2024 (ICML 2024\) and we have already compared our results with those by Yang et al., 2024\. Notably, in Yang et al., 2024 and earlier works on Lyapunov stability such as Wu et al., 2023, Dai et al., 2021, and Chang et al., 2019, they also do not compare with any forward invariant set baselines, since the setting is different.

---

> ### Author Response · Authors · 2024-11-16
> **Rebuttal (3/3): Clarification on branch-and-bound**
>
> ## Clarification on branch-and-bound
>
> >Is the training dataset $\mathbb{D}$ a set of points or regions? What does it mean by $(\underline{x}, \overline{x})$in Line 269?
>
> The training set is a set of regions, not points. Each $(\underline{x}, \overline{x})$ means a subregion (a bounding box).  We defined them at the beginning of Section 3.2 around Line 218-Line 220 in our initial submission:
> “each example $(\underline{\mathbf{x}}^{(k)}, \overline{\mathbf{x}}^{(k)})~(1\leq k\leq n)$ is a subregion in $\mathcal{B}$, defined as a bounding box $\{\mathbf{x}: \mathbf{x}\in\mathbb{R}^{d},\,\underline{\mathbf{x}}^{(k)} \leq \mathbf{x} \leq \overline{\mathbf{x}}^{(k)}\}$ with boundary $\underline{\mathbf{x}}^{(k)}$ and $\overline{\mathbf{x}}^{(k)}$”.
>
> >Does the proposed approach need to verify that g(x)≤0 for every region?
>
> Yes, the condition needs to be verified for the entire region-of-interest $\mathcal{B}$ (i.e., all the subregions). This requirement is necessary for the formal verification of Lyapunov stability and it is the same as that in previous works on Lyapunov stability (Wu et al., 2023; Yang et al., 2024).
>
> >If so, then how does this work scale to high-dimensional systems?
>
> Lyapunov stability is a relatively strong guarantee and existing works on Lyapunov stability with formal guarantees have commonly focused on relatively low-dimensional systems so far (Chang et al., 2019; Wu et al., 2023; Yang et al., 2024), which we have mentioned as a limitation in the conclusion section.
>
> In the conclusion section, we also mentioned that future works may consider training-time branch-and-bound on activation functions instead of input, in order to scale to high-dimensional systems. It is motivated by existing works on neural network verification methods which typically conduct branch-and-bound on activation functions for high-dimensional problems, because there is often sparsity in the active/inactive status of activation functions such as ReLU, which can be efficiently leveraged by verifiers so that branching on activation functions can be more efficient than branching on the large input. However, it remains an open problem to conduct certified training and train verifiable models by training-time branch-and-bound on activation functions, for Lyapunov stability in high-dimensional systems. We believe this is an important direction for future works.

---

### Official Review · Reviewer_ueWG · 2024-10-24

**Soundness:** 3
**Presentation:** 3
**Contribution:** 3
**Rating:** 6
**Confidence:** 5

**Summary:**

The paper aims to train safe neural network controller in dynamic systems in discrete time and continuous action space. The safety is formally verified using lyapunov function (stability) and existing formal NN verifier. The training process consists of recursively splitting the considered input domain (region-of-interest) and learning through backpropagation to fullfill two conditions on each subset: (1) lyapunov stability and (2) controller should not steer outside of input domain. This iteratively increases the region for which stabilty is shown (region of attraction (ROA) due to lyapunov function), which the paper aims to maximize. The approach is demonstrated on three low-dimensional benchmarks.

**Strengths:**

- The paper is very well written, and one can follow along with all arguments and choices nicely.
- The topic of formally verifying the safety of neural network controller is highly relevant and suits ICLR
- The approach is novel and demonstrated on three benchmarks with improvements on related work.
- Even though the approach is simple, it is still effective in achieving its goal.

**Weaknesses:**

Major points
- Approach is based on branch-and-bound, which suffers from the curse of dimensionality and thus might not be applicable in high-dimensional systems. This is also visible in Tab. 3 where the final data set size is already much larger for the 6-dimensional quadrotor benchmark (which probably also increased the runtime). Appendix A also states that they stopped splitting after 5000 training steps for this benchmark. This limitation is mentioned in Sec. 5 but a discussion about the theoretical complexity and directions to overcome this limitation would be helpful.
- The paper only considers the safety property of stability, i.e., the actor should steer to some (predefined?) equilibrium state  (see question below for other safety properties that could be discussed).
- The results in Sec. 4 are only single-dimensional: Over how many seeds are the runs averaged? Please provide stand deviations where applicable, e.g., the area of ROA.

Minor points:
- Eq (2) assumes perfect knowledge of the system with no disturbance (e.g., sensor noise).
- The paper considers only discrete time and continuous action space, which could be highlighted more clearly.
- Spelling / Grammar mistakes: e.g., line 063: "stability condition needs to *be* verified", line 168: "lower bound" should be "upper bound" (?), line 182: "the the", line 183: "an NN" should be "a NN", ...
- Experiments are only done on low-dimensional systems. This limitation is addressed in the paper.
- Convergence to a single point within the region of interest is not well motivated (see question below)
- Appendix A: The considered networks are rather small.
- no repeatability package

Other points that did not directly influence the score
- First contribution: "relatively global guarantee" could be specified more precisely.
- Tab. 1: (re-)explain all variables, e.g., d, n_u are not explained properly.
- Fig. 1: What does the color mean? Is it your Lyapunov function?

**Questions:**

- Using your dynamic splitting, are there certain areas that get splitted more often than other areas? How do these areas differ? E.g., are there more splits at V(x) = 0 or similar? Would be nice to get more insights there. Maybe one can visualize the splitted subsets using a heat map or similar to see in which area the most splits occured.
- Does the number of regions converge to some maximum or do they continuously get split during training?
- Why did you decide to stop splitting after 5000 training steps for the quadrotor benchmark?
- Would it be useful to merge some regions at some point again?
- Is it sensible to assume a single equilibrium state x*? How is x* determined as x* has to be known to evaluate the equations given sin Sec. 3.4? The authors could discuss the implications of this assumption and how the method needs to be adapted for multiple x*.
- Would your approach also work for more equilibrium states / if the system does not converge but should not violate a safety property where the actor should stay outside of some unsafe region after some point in time?
- End of Sec. 3.1: What holds for points x \in B \  S ?
- Addition of adversarial attacks in training objective (6): Why is this necessary? How would training change if this was not included? Would it still work or is it necessary "to get started"? Maybe add to ablation study.
- Sec. 3.3: The initial splits appear a bit random: Would it not be better to start off with the entire region of interest and only refine into subsets where necessary?
- Is it sensible to test each dimension before deciding where to split? This might increase the training time (especially if higher-dimensional systems are considered). Would another heuristic make more sence, e.g., sensitivity?
- Lines 306-309: Isn't this already penalized by the second condition in (4)?

---

> ### Author Response · Authors · 2024-11-23
> **Rebuttal (1/3): branch-and-bound**
>
> We thank the reviewer for identifying our strengths and providing detailed and constructive feedback. We have revised our paper accordingly (with major changes highlighted in blue in the updated PDF), and we address the weakness points and questions below:
>
>
> ## Branch-and-bound
>
> >Approach is based on branch-and-bound, which suffers from the curse of dimensionality and thus might not be applicable in high-dimensional systems... This limitation is mentioned in Sec. 5 but a discussion about the theoretical complexity and directions to overcome this limitation would be helpful.
>
> We think a theoretical analysis for the complexity is still an open problem, as to our knowledge, even existing works on branch-and-bound for test-time verification do not have a theoretical complexity. Given the more complicated nature of training-time branch-and-bound which further involves the training dynamics of neural networks, we believe obtaining a theoretical complexity is highly challenging at this time and thus we have to leave it for future work.
>
> In Section 5, we briefly mentioned directions on supporting high-dimensional systems, potentially by considering splits on activation functions instead of input states. It is motivated by existing works on neural network verification methods which typically conduct branch-and-bound on activation functions for high-dimensional problems, because there is often sparsity in the active/inactive status of activation functions such as ReLU, which can be efficiently leveraged by verifiers so that branching on activation functions can be more efficient than branching on the large input. However, it remains an open problem to conduct certified training and train verifiable models by training-time branch-and-bound on activation functions for high-dimensional systems. We believe this is an important direction for future works.
>
> >Using your dynamic splitting, are there certain areas that get splitted more often than other areas? How do these areas differ? E.g., are there more splits at V(x) \= 0 or similar? Would be nice to get more insights there. Maybe one can visualize the splitted subsets using a heat map or similar to see in which area the most splits occured.
>
> We thank the reviewer for suggesting a visualization for the branch-and-bound. We have added Appendix B for this visualization. As expected, more extensive splits tend to happen when at least one (sometimes all) of the input states is close to that of the equilibrium state, where Lyapunov function values are relatively small and the training tends to be more challenging.
>
> >Does the number of regions converge to some maximum or do they continuously get split during training?
>
> For the relatively easy systems (e.g., inverted pendulum as shown in Table 3), the number of regions naturally converges to some maximum, when CROWN can verify all the regions and thus no more split is needed. If CROWN cannot verify all the regions yet, the number of regions can continue growing, but in our implementation, we early stop the training (when it is already sufficient to verify the model by a more extensive branch-and-bound at test time) or we stop splitting at some point as you mentioned.
>
> If we do not stop splitting, as long as the training can succeed, technically the number of regions can still converge to some maximum ultimately. It is because the test-time branch-and-bound can verify all the models, and if the training-time branch-and-bound achieves a comparable level of splitting, compared to the test-time branch-and-bound, the number of regions can ultimately saturate.
>
> However, we believe it is more reasonable to restrict the number of splits during the training (e.g., by stopping splitting at some point if the number of splits is already enough for successful training and verification), as training-time branch-and-bound is more costly than test-time branch-and-bound (many training epochs may be needed on the regions). It can potentially also make the model work better under a limited number of splits (if the number of splits is already sufficient) so that it may reduce the number of required splits at test time.
>
> >Why did you decide to stop splitting after 5000 training steps for the quadrotor benchmark?
>
> We have discussed our motivation above. Below we compare the performance when the early stopping for the splitting is enabled v.s. disabled. The training could work under both settings which produces similar ROA, while early stopping the splitting achieves a slightly lower verification time cost at test time. The difference is overall small, so it is optional to stop the splitting early here. We did not early stop the splitting for other systems in our experiments because the training could finish in fewer than 5000 steps.
>
> | Stopping the splitting after 5000 training steps | Time | ROA |
> | :---- | :---- | :---- |
> | Enabled | 11.5min | 54.39 |
> | Disabled | 13.0min | 54.70 |

---

> ### Author Response · Authors · 2024-11-23
> **Rebuttal (2/3): branch-and-bound (cont.); safety properties; multiple random seeds**
>
> ## Branch-and-bound (cont.)
>
> >Would it be useful to merge some regions at some point again?
>
> Thanks for the great suggestion. We agree that it can be potentially useful to merge regions -- we may track the branch-and-bound tree during the training, and for two subregions coming from a bigger parent region, if both of them can be verified, we may merge them into the parent region again. In our future work, we plan to investigate the scalability and new techniques to handle higher-dimensional systems, and we will try to implement this merging strategy to see if it can help for harder systems.
>
> >Is it sensible to test each dimension before deciding where to split? This might increase the training time (especially if higher-dimensional systems are considered). Would another heuristic make more sence, e.g., sensitivity?
>
> Testing each dimension is a relatively simple approach and it can indeed be more costly if more branching is needed especially for higher-dimensional systems. We would agree that future works may design a smart and efficient heuristic when trying to support higher-dimensional systems.
>
> >Sec. 3.3: The initial splits appear a bit random: Would it not be better to start off with the entire region of interest and only refine into subsets where necessary?
>
> We used an initial split instead of starting with a single region, in order to have enough initial regions to fill $1\sim 2$ batches according to the batch size, so that the batch size can remain stable during the training. If we start with a single region, then the actual batch size would be very small and unstable in the beginning (starting from from batch size=1), which we think may not be good for the optimizer, as existing deep learning works typically have a fixed batch size. We have revised our paper and added an explanation.
>
> ## Applicability to other safety properties
>
> >Convergence to a single point within the region of interest is not well motivated (see question below).
>
> >Is it sensible to assume a single equilibrium state x*? How is x* determined as x* has to be known to evaluate the equations given sin Sec. 3.4? The authors could discuss the implications of this assumption and how the method needs to be adapted for multiple x*.
>
> It is a typical setting in Lyapunov asymptotic stability. We followed previous works (Wu et al., 2023; Yang et al., 2024) to consider dynamical systems with a single equilibrium state $x^*$. These systems are already known to have a single equilibrium state (all at 0 here), according to previous works or textbooks (mentioned in Section 4.1), and thus we treat it as a prior knowledge. We have revised our paper and added a sentence in Section 3.1 to clarify that.
>
> >Would your approach also work for more equilibrium states
>
> According to textbook Murray et al., 2017 (Section 4.1 in the book), if there are multiple equilibrium points, we will need to study the Lyapunov asymptotic stability w.r.t. each equilibrium point individually. And then our method can be applied for each equilibrium point independently.
>
> Murray, R. M., Li, Z., & Sastry, S. S. (2017). A mathematical introduction to robotic manipulation. CRC press.
>
> >The paper only considers the safety property of stability, i.e., the actor should steer to some (predefined?) equilibrium state (see question below for other safety properties that could be discussed).
> >if the system does not converge but should not violate a safety property where the actor should stay outside of some unsafe region after some point in time?
>
> Our new training framework is generally formulated in Section 3.2 and Section 3.3, with a specific instantiation and focus for Lyapunov asymptotic stability in Section 3.4. We believe that our framework also has the potential for broader applications including other safety properties in control, such as reachability or forward invariance to ensure that a controller can stay outside of some unsafe region, or systems with disturbance, as you mentioned.  These are interesting future directions.
>
> Many previous works focus on a particular kind of safety property -- e.g., Dai et al. 2021, Wu et al. 2023, Yang et al. 2024 all focused on Lyapunov asymptotic stability. Thus we believe it is reasonable for us to also focus on Lyapunov asymptotic stability in this paper.
>
> ## Random seeds
>
> >The results in Sec. 4 are only single-dimensional: Over how many seeds are the runs averaged? Please provide stand deviations where applicable, e.g., the area of ROA.
>
> We followed the previous work Yang et al., 2024 to use a single seed, and thus standard deviations were not originally applicable. Additionally, we have extended our experiments to use 5 different random seeds on the 2D quadrotor systems as shown below. Our method has relatively stable performance while significantly outperforming Yang et al., 2024.
>
> | Method | Time | ROA |
> | :---- | :---- | :---- |
> | Yang et al., 2024 (single seed) | 1.1 hrs | 3.29 |
> | Ours (5 seeds) | 8.27±1.70 min | 46.77±5.26 |

---

> > ### Author Response · Authors · 2024-11-23
> > **Rebuttal (3/3): minor issues or questions**
> >
> > ## Minor issues and questions
> >
> > >The paper considers only discrete time and continuous action space, which could be highlighted more clearly.
> >
> > We have mentioned “discrete-time” for multiple times when we mention the problem, e.g., “learning and verifying Lyapunov-stable neural controllers in discrete-time nonlinear dynamical systems”. We have revised our Section 3.1 (in the “Specifications for Lyapunov-stable neural control” paragraph) to highlight that the control action is continuous.
> >
> > >First contribution: "relatively global guarantee" could be specified more precisely.
> >
> > We had a “which” clause to explain “relatively global guarantees” as “which provably hold on the entire input region-of-interest”. We have extended it to say “which provably hold on the entire input region-of-interest instead of only small local regions around a finite number of data points”.
> >
> > >Tab. 1: (re-)explain all variables, e.g., d, n_u are not explained properly.
> > We have revised the paper to explain $d$ and $n_u$ in the table caption.
> >
> > >Fig. 1: What does the color mean? Is it your Lyapunov function?
> > Yes, it denotes the value of the learned Lyapunov function. We have revised our caption.
> >
> > >Lines 306-309: Isn't this already penalized by the second condition in (4)?
> >
> > (4) aims to achieve $V(\mathbf{x}_{t+1})-V(\mathbf{x}_t) \leq -\kappa V(\mathbf{x}_t)$, while the construction in Lines 306-309 (in our initial submission before revision) aims to guarantee $V(\mathbf{x}^*)=0$ and $ V(\mathbf{x})>0~(\forall \mathbf{x}\neq\mathbf{x}^*)$ by construction. Both of these two are needed as indicated in (3).
> >
> > >End of Sec. 3.1: What holds for points x \in B \ S ?
> >
> > $\mathcal{B}$ is the region of interest while $\mathcal{S}$ is the ROA. $g(\mathbf{x})\leq 0$ should hold for $\mathbf{x}\in\mathcal{B}$, which can then guarantee that  Eq. (3) (i.e., the Lyapunov condition) holds for states $x\in\mathcal{S}$.
> >
> > >Addition of adversarial attacks in training objective (6): Why is this necessary? How would training change if this was not included? Would it still work or is it necessary "to get started"? Maybe add to ablation study.
> >
> > Since learning an empirically stable controller is easier than learning a verifiably stable one, adding an adversarial attack objective can help the training more quickly reach a (roughly) empirically stable controller with most counterexamples eliminated, so that certified training can better focus on making the model verifiable. Without this objective, we find that the training struggles to find an empirically stable controller. We conducted an ablation study on the 2D quadrotor system. If we do not add the adversarial attack objective, after 10000 training steps (in our default setting, the model can already be fully verified after 10000 training steps), around 18% of the regions still have counterexamples as found by adversarial attacks. Additionally, the adversarial attack objective also helps to achieve that at least no counterexample can be empirically found, even if verified bounds by CROWN and IBP cannot yet verify all the examples in the current dataset $(\underline{\mathbf{x}}, \overline{\mathbf{x}})\in \mathbb{D}$, as we may still be able to fully verify Eq. (1) at test time using a stronger verifier enhanced with large-scale branch-and-bound. We have revised our paper to more clearly explain our motivation of adding the adversarial attack objective Eq. (6).
> >
> > >no repeatability package
> >
> > We will release our code upon publication.
> >
> > We have also fixed the typos you pointed out.

---

> > > ### Comment · Reviewer_ueWG · 2024-11-25
> > >
> > > Thank you very much for your clarifications and additional experiments. Overall, I find the direction very promising. I think it would be best for the paper to investigate the future directions discussed in this thread and also raised by other reviewers.

---

> > > > ### Author Response · Authors · 2024-11-30
> > > > **Thanks for acknowledging our rebuttal**
> > > >
> > > > We would like to thank Reviewer ueWG for the positive review and acknowledging our rebuttal. We will continue investigating the certified training with training-time branch-and-bound direction and its broader applications in our future research.

---

### Official Review · Reviewer_SSH5 · 2024-11-02

**Soundness:** 2
**Presentation:** 3
**Contribution:** 2
**Rating:** 5
**Confidence:** 4

**Summary:**

The paper proposes a new certified training framework for generating neural networks with relative global guarantees. By introducing a training-time branch-and-bound method that dynamically maintains a training dataset, the most difficult sub-regions are iteratively divided into smaller sub-regions. The verification boundaries of these sub-regions can be computed more tightly to simplify training, addressing the challenges of certified training in relatively large input regions.

**Strengths:**

This paper is well-written and easy to follow with clear logic and a well-structured layout. The comparative experimental results, along with accompanying visualizations, demonstrate the effectiveness of the proposed method, and the necessity of the training dataset is highlighted through ablation experiments.

**Weaknesses:**

Although this paper presents valuable insights, there are still several areas that need improvement.

It is well-known that the synthesis speed of Lyapunov functions for low-dimensional nonlinear systems is very fast, especially with learning methods based on SMT solvers and counterexample-guided approaches. These methods not only provide formal correctness guarantees (in contrast to the simulation testing used in this paper) but also leverage efficient neural network architectures, demonstrating strong learning capabilities. However, the certified training framework proposed in this paper, based on the branch-and-bound idea, lacks theoretical support and does not discuss the soundness of the proposed method.

The experimental section of this paper provides only limited comparisons with the related work of Yang et al., 2024, making it difficult to assess the reliability of the experimental results. The authors should also compare their method with other approaches to highlight the contributions of this work. Additionally, we would like to see more examples involving high-dimensional systems to demonstrate the efficiency of the proposed method.

Due to the limited length of the article, the discussion of related work should be appropriately integrated into the introduction. It is necessary to introduce the relevant theories and background on system stability and Lyapunov functions. The paper focuses on the work of Yang et al.; we suggest that the authors provide a brief explanation to highlight the differences between their method and the new approach presented in this paper.

**Questions:**

I would like to emphasize that this paper is well-written and clear. But there are some questions and uncertainties that I hope the authors can kindly address.

1.	Learning-based methods often lack interpretability. Although a Lyapunov function is obtained through learning, there may be some errors. Does the learned Lyapunov function truly satisfy the stability conditions? Could the authors provide an explanation for this?

2.	The authors state in the paper, "Empirically, the neural controllers generated by the training framework in this work can be verified to satisfy the Lyapunov conditions, with a larger region of attraction (ROA), and the Lyapunov conditions can be verified more effectively during testing." Generally, the consistency between experimental results and theoretical foundations provides assurance for the validity of the method. I do not understand why the method's efficiency is indicated solely based on empirical evidence, especially since these experiments are limited and conducted in low dimensions.

---

> ### Author Response · Authors · 2024-11-20
> **Rebuttal (1/2): clarification on soundness guarantees; additional baseline added**
>
> We thank the reviewer for constructive feedback and we have revised our paper accordingly (see our updated PDF, with changes highlighted in blue). We believe there was some misunderstanding regarding the soundness of our method (the models produced by our method have actually been formally verified) and we have provided clarification below. We also explained our comparison with baselines and extended our comparison to include an earlier baseline. We also explained the common limitation of existing works in the dimension of systems and provided directions for future works. Finally, we also added an early discussion on the background of Lyapunov asymptotic stability in the introduction section.
>
> ## Soundness guarantees
>
> We would like to clarify that **our method produces models with formal soundness guarantees**.
>
> As mentioned in the “Implementation” paragraph in Section 4.1, after the models are trained, we use α,β-CROWN, a formal complete verifier, to verify the Lyapunov condition with ROA for our models. This evaluation with formal verification follows Yang et al., 2024 and thus it has provided the soundness for our work. All the models can be successfully verified as shown in Table 2 and Table 3\. We have also revised our paper (near the end of Section 3.2) to clarify that a formal verifier is employed at test time to ensure soundness.
>
> >Although a Lyapunov function is obtained through learning, there may be some errors. Does the learned Lyapunov function truly satisfy the stability conditions? Could the authors provide an explanation for this?
>
> As explained above, the conditions have been formally verified at test time by α,β-CROWN.
>
> >These methods not only provide formal correctness guarantees (in contrast to the simulation testing used in this paper)
>
> We would like to clarify that our paper does not use “simulation testing”. Instead, we use formal verification by α,β-CROWN at test time, and thus our method achieves formal guarantees.
>
> >Generally, the consistency between experimental results and theoretical foundations provides assurance for the validity of the method. I do not understand why the method's efficiency is indicated solely based on empirical evidence, especially since these experiments are limited and conducted in low dimensions.
>
> The performance of neural network-based approaches typically need to be demonstrated by experiments. Although there are formal soundness guarantees in our evaluation after models are trained, it is hard to theoretically guarantee what solution the training process can find, just like most other deep learning works, due to the complicated nature of the training process. This is consistent with previous works such as Dai et al., 2021, Wu et al., 2023, Yang et al., 2024 \-- none of them could provide any guarantee on the convergence of training, but the Lyapunov conditions can be formally verified at test time after models are trained in the experiments.
>
> The empirical advantage of our proposed method is also supported by our motivations \-- since we use verified bounds at test time for verification, we propose to optimize verified bounds during training by certified training, and a training-time branch-and-bound is proposed to enhance the training given that the properties need to satisfy on the entire region-of-interest which is a relatively large region.
>
> ## Comparison with baselines
>
> >The experimental section of this paper provides only limited comparisons with the related work of Yang et al., 2024, making it difficult to assess the reliability of the experimental results. The authors should also compare their method with other approaches to highlight the contributions of this work.
>
> Yang et al., 2024 is the previous state-of-the-art work on this problem and thus we mainly compared our work with Yang et al., 2024 to demonstrate our performance. There are a limited number of learning-based approaches on the same problem setting. Additionally, we have revised our paper to also include applicable results for Wu et al. 2023 in Table 2\. Wu et al. 2023 underperforms Yang et al., 2024 with much smaller ROA and is only applicable on some of the systems (e.g., Wu et al., 2023 cannot correctly scale to systems with 6 input states such as the 2D quadrotor system, as discussed in Yang et al., 2024). Our method also achieves much larger ROA compared to Wu et al., 2023.

---

> ### Author Response · Authors · 2024-11-20
> **Rebuttal (2/2): high-dimensional systems; improved introduction section**
>
> ## High-dimensional systems
>
> >Additionally, we would like to see more examples involving high-dimensional systems to demonstrate the efficiency of the proposed method.
>
> As acknowledged in our conclusion section, existing works and our work so far are all limited to relatively low-dimensional systems, for the problem of Lyapunov (asymptotic) stability which is a relatively strong guarantee.
>
> In the conclusion section, we also mentioned that future works may consider training-time branch-and-bound on activation functions instead of input, in order to scale to high-dimensional systems. It is motivated by existing works on neural network verification methods which typically conduct branch-and-bound on activation functions for high-dimensional problems, because there is often sparsity in the active/inactive status of activation functions such as ReLU, which can be efficiently leveraged by verifiers so that branching on activation functions can be more efficient than branching on the large input. However, it remains an open problem to conduct certified training and train verifiable models by training-time branch-and-bound on activation functions, for Lyapunov stability in high-dimensional systems. We believe this is an important direction for future works.
>
> ## Background of Lyapunov stability and difference with Yang et al., 2024
>
> >Due to the limited length of the article, the discussion of related work should be appropriately integrated into the introduction. It is necessary to introduce the relevant theories and background on system stability and Lyapunov functions. The paper focuses on the work of Yang et al.; we suggest that the authors provide a brief explanation to highlight the differences between their method and the new approach presented in this paper.
>
> We thank the reviewer for suggesting an explanation on the background of Lyapunov asymptotic stability, which we agree is important. We have revised our introduction section to include an explanation (highlighted in blue in the updated PDF), as:
>
> "It involves finding a Lyapunov function which intuitively characterizes the energy of input states, where the global minima of Lyapunov function is at an equilibrium point. If it can be guaranteed that for any state within a region-of-attraction (ROA), the controller always makes the system evolve towards states with lower Lyapunov function values, then it implies that starting from any state within the ROA, the controller can always make the system converge towards the equilibrium point and thus the stability can be guaranteed."
>
> We have also highlighted the difference compared to previous works in the revised introduction section, as:
>
> *"To do this, we optimize
> for verified bounds on subregions of inputs instead of only violations on concrete counterexample
> data points, and thus our approach differs significantly compared to Wu et al. (2023); Yang et al.
> (2024)."*

---

> > ### Comment · Reviewer_SSH5 · 2024-11-26
> >
> > Thank you for your detailed response and promise to clarify the points in the revision. I don't have further questions. (We look forward to the authors considering branch-and-bound strategies for activation functions during the training process, in order to extend this approach to high-dimensional systems.)

---

> > > ### Author Response · Authors · 2024-11-30
> > > **Thanks for acknowledging our rebuttal and a reminder to reconsider rating**
> > >
> > > We thank the reviewer for acknowledging our rebuttal. We would like to note that the revision mentioned in our last rebuttal had already been integrated into the updated PDF (i.e., it was not just a "promise to clarify the points in the revision"). Additionally, given that our rebuttal seems to have addressed the weakness points in your review, we would like to remind you to kindly consider updating your rating, as we are approaching the end of the discussion period.Thanks.

---

### Official Review · Reviewer_pcDF · 2024-11-04

**Soundness:** 3
**Presentation:** 3
**Contribution:** 2
**Rating:** 5
**Confidence:** 4

**Summary:**

This paper introduces a verification framework for certified training based on BaB techniques, and conducts a case study on the neural Lyapunov control task. Unlike adversarial training techniques widely used in the literature, this paper obtains a relatively global output guarantee without using time consuming verifiers such as SMT, MIP, etc.

**Strengths:**

1. Proposed method verifies the condition $g_\theta(x)\leq 0$ for $x\in\mathcal{B}$ instead of adversarial examples.
2. The experiments on Mujoco environments, particularly the significant reduction in verification time and the expansion of the region of attraction (ROA), provide strong evidence for the framework's effectiveness. Achieving a 5X faster verification time and a 16X larger ROA for the 2D quadrotor system demonstrates impactful results.

**Weaknesses:**

1. Line 215: typo “on the entire”.
2. Relaxation of the activation function is mainly RELU based. It would be more interesting to see some other activation functions.
3. It is claimed in the introduction that “this approach supports the random initialization” in lines 324 to 330. It would be great to have some experiments with different randomized initialization and the initialization impacts on verification and ROA calculations.
4. In line 307, it would be great to explain more why replacing margin $\rho$ to $\rho+\epsilon$ will prevent the controller going out of ROI.
5. The setting and Lyapunov synthesis condition is exactly the same as [1]. The only difference is that the certification uses BaB framework instead of adversarial robustness, which makes the contribution not too significant, since BaB framework for neural network verification is also pretty common (e.g., [7,8]), especially for RELU network. It would be great to see some variations using different reachability tools other than $\alpha-\beta$ CROWN, such as in Sherlock [2], nnenum [3], etc.
6. Also it would be interesting to compare the current setup with some existing NNCS (Neural Network Control System) verification tools, such as NNV [4], Polar-Express [5], CORA [6], etc.
7. Though not limited to this method, all the similar methods seem to have a bottleneck on dimensionality.

References:
1. Yang, Lujie, et al. "Lyapunov-stable Neural Control for State and Output Feedback: A Novel Formulation." Forty-first International Conference on Machine Learning.
2. Dutta, S., Chen, X., Jha, S., Sankaranarayanan, S., Tiwari, A.: Sherlock-a tool for verification of neural network feedback systems: demo abstract. In: Proceedings of the 22nd ACM International Conference on Hybrid Systems: Computation and Control. pp. 262–263 (2019)
3. Bak, Stanley. "nnenum: Verification of relu neural networks with optimized abstraction refinement." NASA formal methods symposium. Cham: Springer International Publishing, 2021.
4. Lopez, D.M., Choi, S.W., Tran, H.D., Johnson, T.T.: Nnv 2.0: the neural network verification tool. In: International Conference on Computer Aided Verification. pp. 397–412. Springer (2023)
5. Wang, Y., Zhou, W., Fan, J., Wang, Z., Li, J., Chen, X., Huang, C., Li, W., Zhu, Q.: Polar-express: Efficient and precise formal reachability analysis of neural-network controlled systems. IEEE Transactions on Computer-Aided Design of Integrated Circuits and Systems (2023)
6. Althoff, Matthias, and Niklas Kochdumper. "CORA 2016 manual." TU Munich 85748 (2016).
7. Bunel, Rudy, et al. "Branch and bound for piecewise linear neural network verification." Journal of Machine Learning Research 21.42 (2020): 1-39.
8. Wang, Shiqi, et al. "Beta-crown: Efficient bound propagation with per-neuron split constraints for neural network robustness verification." Advances in Neural Information Processing Systems 34 (2021): 29909-29921.

**Questions:**

See above in weaknesses

---

> ### Author Response · Authors · 2024-11-21
> **Rebuttal (1/2): summary of rebuttal; our contributions on training; comparison between verifiers**
>
> We thank the reviewer for constructive feedback. We have clarified our contributions compared to previous works, where we highlight our novel contributions on training, not verification. We have extended our related work section to include all the references you mentioned. We also added additional results on varying the activation function and random initialization. Finally, we provided some additional explanation and fixed a typo.
>
> ## Comparison with previous works
>
> >The setting and Lyapunov synthesis condition is exactly the same as \[1\]. The only difference is that the certification uses BaB framework instead of adversarial robustness, which makes the contribution not too significant, since BaB framework for neural network verification is also pretty common (e.g., \[7,8\]), especially for RELU network.
>
> We would like to clarify that our main contribution is on **the first certified training framework for Lyapunov-stable control**, where our training framework is enhanced with **training-time branch-and-bound.**  Although our problem setting follows Yang et al., 2024, **our focus is on the training framework which we believe is novel**. Our contributions on the training framework are also significantly different compared to Bunel et al., 2020, Wang et al., 2021 which are for verifying trained models. Importantly, our method could enable the training of more verification-friendly models to obtain stronger guarantees (such as larger ROA in this paper), which previous works such as Bunel et al., 2020, Wang et al., 2021 could not do. Our contributions are thus crucial for training/building verifiable models in mission-critical applications, not just verifying/testing existing models.
>
> >It would be great to see some variations using different reachability tools other than α−β CROWN, such as in Sherlock \[2\], nnenum \[3\], etc. Also it would be interesting to compare the current setup with some existing NNCS (Neural Network Control System) verification tools, such as NNV \[4\], Polar-Express \[5\], CORA \[6\], etc.
>
> We have extended our related work section to cover all the references you mentioned.
>
> Verification for Lyapunov-stable neural control has been benchmarked in the recent 5th International Verification of Neural Networks Competition (VNN-COMP'24) ([https://sites.google.com/view/vnn2024](https://sites.google.com/view/vnn2024)). Models were developed by Yang et al., 2024 and built into a benchmark called LSNC (short for Lyapunov-stable neuron control). The results have been reported at [https://docs.google.com/presentation/d/1RvZWeAdTfRC3bNtCqt84O6IIPoJBnF4jnsEvhTTxsPE/edit\#slide=id.g279a3ebee4e\_5\_383](https://docs.google.com/presentation/d/1RvZWeAdTfRC3bNtCqt84O6IIPoJBnF4jnsEvhTTxsPE/edit#slide=id.g279a3ebee4e_5_383) (publicly available, linked at [https://sites.google.com/view/vnn2024](https://sites.google.com/view/vnn2024)).
>
> The competition has participants including α−β-CROWN, NNV, nnenum, and CORA which you have mentioned. Among those participants, only α−β-CROWN could support the LSNC benchmark. In total, there were only two teams of participants supporting the LSNC benchmark (α−β-CROWN and PyRAT), with α−β-CROWN significantly outperforming PyRAT (α−β-CROWN successfully verified all, while PyRAT only verified 15 out of 40 subregions). Therefore, VNN-COMP’24 has already shown the difference of those verifiers in terms of verifying trained models. Since our focus is on training, not verification, we believe it is reasonable for us to adopt the state-of-the-art verifier on this problem (i.e., α−β-CROWN) for verification at test time. Additionally, we would also like to clarify that Lyapunov (asymptotic) stability (with a guarantee on convergence towards the equilibrium under an infinite time horizon) is different from reachability (with finite time) handled by some tools such as Sherlock, Polar-Express, etc.

---

> > ### Author Response · Authors · 2024-11-21
> > **Rebuttal (2/2): activation function; initialization; additional explanation and typo fixes**
> >
> > ## Activation function
> >
> > >Relaxation of the activation function is mainly RELU based. It would be more interesting to see some other activation functions.
> >
> > Our work not only has ReLU but also Leaky ReLU. As mentioned in Appendix A, models with NN Lyapunov functions have Leakly ReLU activation functions. The activation functions and model architectures in our experiments follow Yang et al., 2024\. Previous certified training works also mainly used ReLU activations.
> >
> > We have tried using Sigmoid activation for the 2D quadrotor system, as our training framework is general to support other activation functions. We find that Sigmoid activation achieves a similar ROA (55.57 v.s. 54.39 when comparing Sigmoid v.s. ReLU) but the time of verification at test time is larger (23.7min v.s. 11.5min when comparing Sigmoid v.s. ReLU). Overall, we think it is more suitable to keep using ReLU, but this experiment has demonstrated the applicability of our training framework on activation functions which are not piecewise linear.
> >
> > | Activation function | Time | ROA |
> > | :---- | :---- | :---- |
> > | ReLU  | 11.5min | 54.39 |
> > | Sigmoid | 23.7min | 55.57 |
> >
> > ## Initialization
> >
> > >It is claimed in the introduction that “this approach supports the random initialization” in lines 324 to 330\. It would be great to have some experiments with different randomized initialization and the initialization impacts on verification and ROA calculations.
> >
> > We previously used the default Xavier initialization (Glorot & Bengio, 2010) which is the default choice in PyTorch. We have added an experiment to compare the Xavier initialization with another well-known initialization method, Kaiming initialization (He et al., 2015). Both two initialization methods achieve similar ROA but Kaiming initialization archives a shorter verification time. This experiment demonstrates the effectiveness of our method when the random initialization method is varied, and users may potentially use Kaiming initialization for training.
> >
> > | Initialization | Time | ROA |
> > | :---- | :---- | :---- |
> > | Xavier (default in PyTorch)  | 11.5min | 54.39 |
> > | Kaiming | 8.6min | 53.81 |
> >
> > We would like to clarify that by mentioning “random initialization” in our paper, it is mainly relative to previous works which used a specialized initialization from linear quadratic regulator (LQR) (by first training the model to fit the LQR solution), while we used the default weight initialization provided by PyTorch *to remove the burden of using a traditional method (e.g., LQR) before the training*. It doesn’t mean that the initialization can be arbitrary. Instead, we would recommend following the common practice in deep learning for initializing the parameters.
> >
> > Glorot, X., & Bengio, Y. (2010). Understanding the difficulty of training deep feedforward neural networks. In Proceedings of the thirteenth international conference on artificial intelligence and statistics (pp. 249-256). JMLR Workshop and Conference Proceedings.
> >
> > He, K., Zhang, X., Ren, S., & Sun, J. (2015). Delving deep into rectifiers: Surpassing human-level performance on imagenet classification. In Proceedings of the IEEE international conference on computer vision (pp. 1026-1034).
> >
> > ## Preventing the controller from going out of ROI
> >
> > >In line 307, it would be great to explain more why replacing margin ρ to ρ+ϵ will prevent the controller going out of ROI.
> >
> > We would like to clarify that the change is not “replacing margin ρ to ρ+ϵ”. Instead, our change is adding the constraint $V(\\mathbf{x}\_{t+1})\\geq \\rho+\\epsilon $ for $\\mathbf{x}\_{t+1}\\notin\\mathcal{B}$ (in contrast to not adding the constraint).
> >
> > Intuitively, this means that any state out of $\\mathcal{B}$ (ROI) should have a Lyapunov function value greater than the sublevel set threshold $\\rho$ (i.e., no smaller than $\\rho+\\epsilon$ with a small margin $\\epsilon$).
> >
> > We have revised our paper and added an explanation to clarify that the constraint is used to:
> >
> > “prevent wrongly minimizing the violation by going out of the region-of-interest as $\mathbf{x}\_{t+1}\\notin\\mathcal{B}$ while making $ V(\mathbf{x}\_{t+1})\~(\mathbf{x}\_{t+1}\\notin\\mathcal{B})$ small, such that the violation $V(\mathbf{x}\_{t+1})-(1-\\kappa)V(\mathbf{x}\_t)$ appears to be small yet missing the $\mathbf{x}\_{t+1}\\in\\mathcal{B}$ requirement.”
> >
> > ## Typo
> >
> > Finally, we thank the reviewer for spotting a typo and we have fixed it.

---

> > > ### Comment · Reviewer_pcDF · 2024-11-30
> > >
> > > Thank you for addressing my comments and providing additional experiments. I have also read through the comments from other reviewers and the responses provided. I think training more verification-friendly models is indeed a promising direction to combat the verification challenges in practical systems, although the proposed BnB approach is a bit limited in depth and novelty. I would encourage the authors to continue working on this direction, refine the approach, and provide a more comprehensive study (e.g, applying such approach in various verification tasks).

---

> ### Author Response · Authors · 2024-11-30
> **Thanks for acknowledging our rebuttal and reminder to reconsider rating**
>
> We thank the reviewer for acknowledging our rebuttal. We agree that this is a promising direction and we will continue investigating the certified training with training-time branch-and-bound direction and its broader applications in our future research. However, we believe that we have already addresssed the weakness points raised in the initial review. We do acknowledge that this work has limitations which, however, are common limitations which exist in other papers recently published in similar venues (the focus on Lyapunov stability and the limitation to relatively low-dimensional systems are common in previous works including Wu et al., 2023 in NeurIPS 2023 and Yang et al., 2024 in ICML 2024). Therefore, we would like to gentlely request the reviewer to reconsider your rating. Thanks.

---

### Note · Authors · 2024-12-02

I have read and agree with the venue's withdrawal policy on behalf of myself and my co-authors.